# Successor Feature Landmarks for Long-Horizon Goal-Conditioned Reinforcement Learning

**Christopher Hoang** [1] **Sungryull Sohn** [1,2] **Jongwook Choi** [1]
**Wilka Carvalho** [1] **Honglak Lee** [1,2]
[1]University of Michigan [2]LG AI Research
{choang, srsohn, jwook, wcarvalh, honglak}@umich.edu

## Abstract

Operating in the real-world often requires agents to learn about a complex environment and apply this understanding to achieve a breadth of goals. This problem, known as goal-conditioned reinforcement learning (GCRL), becomes especially challenging for long-horizon goals. Current methods have tackled this problem by augmenting goal-conditioned policies with graph-based planning algorithms. However, they struggle to scale to large, high-dimensional state spaces and assume access to exploration mechanisms for efficiently collecting training data. In this work, we introduce Successor Feature Landmarks (SFL), a framework for exploring large, high-dimensional environments so as to obtain a policy that is proficient for any goal. SFL leverages the ability of successor features (SF) to capture transition dynamics, using it to drive exploration by estimating state-novelty and to enable high-level planning by abstracting the state-space as a non-parametric landmark-based graph. We further exploit SF to directly compute a goal-conditioned policy for inter-landmark traversal, which we use to execute plans to "frontier" landmarks at the edge of the explored state space. We show in our experiments on MiniGrid and ViZDoom that SFL enables efficient exploration of large, high-dimensional state spaces and outperforms state-of-the-art baselines on long-horizon GCRL tasks[1].

## 1 Introduction

Consider deploying a self-driving car to a new city. To be practical, the car should be able to explore the city such that it can learn to traverse from any starting location to any destination, since the destination may vary depending on the passenger. In the context of reinforcement learning (RL), this problem is known as goal-conditioned RL (GCRL) [12, 13]. Previous works [31, 1, 24, 26, 17] have tackled this problem by learning a goal-conditioned policy (or value function) applicable to any reward function or "goal." However, the goal-conditioned policy often fails to scale to long-horizon goals [11] since the space of state-goal pairs grows intractably large over the horizon of the goal.

To address this challenge, the agent needs to (a) explore the state-goal space such that it is proficient for any state-goal pair it might observe during test time and (b) reduce the effective goal horizon for the policy learning to be tractable. Recent work [25, 11] has tackled long-horizon GCRL by leveraging model-based approaches to form plans consisting of lower temporal-resolution subgoals. The policy is then only required to operate for short horizons between these subgoals. One line of work learned a universal value function approximator (UVFA) [31] to make local policy decisions and to estimate distances used for building a landmark-based map, but assumed a low-dimensional state space where the proximity between the state and goal could be computed by the Euclidean distance [11]. Another line of research focused on visual navigation tasks conducted planning over

---

[1]The demo video and code can be found at https://2016choang.github.io/sfl.

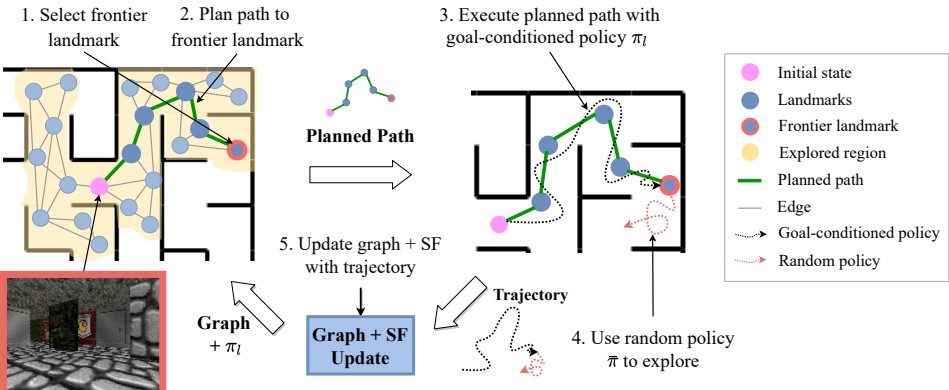

Figure 1: High-level overview of SFL. **1.** During *exploration*, select a frontier landmark (red circled dot) lying at the edge of the explored region as the target goal. During *evaluation* (not shown in the figure), the actual goal is selected as the target goal. **2.** Use the graph to plan a landmark path (green lines) to the target goal. **3.** Execute the planned path with the goal-conditioned policy (black dotted arrow). **4.** During *exploration*, upon reaching the frontier landmark, deploy the random policy (red dotted arrow) to reach novel states in unexplored areas. **5.** Use each transition in the trajectory to update the graph and SF (see Figure 2). **Note:** The agent is never shown the top-down view of the maze and only uses first-person image observations (example on left) to carry out these steps. Goals are also given as first-person images.

graph representations of the environment [29, 9, 16, 4]. However, these studies largely ignored the inherent exploration challenge present for large state spaces, and either assumed the availability of human demonstrations of exploring the state space [29], the ability to spawn uniformly over the state space [9, 16], or the availability of ground-truth map information [4].

In this work, we aim to learn an agent that can tackle long-horizon GCRL tasks and address the associated challenges in exploration. Our key idea is to use successor features (SF) [15, 2] — a representation that captures transition dynamics — to define a novel distance metric, Successor Feature Similarity (SFS). First, we exploit the transfer ability of SF to formulate a goal-conditioned value function in terms of SFS between the current state and goal state. By just learning SF via self-supervised representation learning, we can directly obtain a goal-conditioned policy from SFS without any additional policy learning. Second, we leverage SFS to build a landmark-based graph representation of the environment; the agent adds observed states as landmarks based on their SFS-predicted novelty and forms edges between landmarks by using SFS as a distance estimate. SF as an abstraction of transition dynamics is a natural solution for building this graph when we consider the MDP as a directed graph of states (nodes) and transitions (edges) following [11]. We use this graph to systematically explore the environment by planning paths towards landmarks at the "frontier" of the explored state space and executing each segment of these planned paths with the goal-conditioned policy. In evaluation, we similarly plan and execute paths towards (long-horizon) goals. We call this framework *Successor Feature Landmarks* (SFL), illustrated in Figure 1.

Our contributions are as follows: (i) We use a single self-supervised learning component that captures dynamics information, SF, to build all the components of a graph-based planning framework, SFL. (ii) We claim that this construction enables knowledge sharing between each module of the framework and stabilizes the overall learning. (iii) We introduce the SFS metric, which serves as a distance estimate and enables the computation of a goal-conditioned Q-value function *without further learning*. We evaluate SFL against current graph-based methods in long-horizon goal-reaching RL and visual navigation on **MiniGrid** [6], a 2D gridworld, and **ViZDoom** [37], a visual 3D first-person view environment with large mazes. We observe that SFL outperforms state-of-the-art navigation baselines, most notably when goals are furthest away. In a setting where exploration is needed to collect training experience, SFL significantly outperforms the other methods which struggle to scale in **ViZDoom**'s high-dimensional state space.

## 2 Related Work

**Goal-conditioned RL**. Prior work has tackled GCRL by proposing variants of goal-conditioned value functions such as UVFAs which estimate cumulative reward for any given state-goal pair [23, 34, 31, 26]. HER [1] improved the sample efficiency in training UVFAs by relabeling reached states as goals. Mapping State Space (MSS) [11] then extended UVFAs to long-horizon tasks by using a UVFA as both a goal-conditioned policy and a distance metric to build a graph for high-level planning. MSS also addressed exploration by selecting graph nodes to be at edge of the map's explored region via farthest point sampling. However, this method was only evaluated in low-dimensional state spaces. LEAP [25] used goal-conditioned value functions to form and execute plans over latent subgoals, but largely ignored the exploration question. Conversely, other works [27, 5] have worked on exploration for goal-reaching policies, but do not tackle the long-horizon case. ARC [10] proposed learning representations that measure state similarity according to the output of a maximum entropy goal-conditioned policy, which can be utilized towards exploration and long-horizon hierarchical RL. However, ARC assumes access to the goal-conditioned policy, which can be difficult to obtain in large-scale environments. Our method can achieve both efficient exploration and long-horizon goal-reaching in high-dimensional state spaces with a SF-based metric that acts as a goal-conditioned value function and distance estimate for graph-building.

**Graph-based planning.** Recent approaches have tackled long-horizon tasks, often in the context of visual navigation, by conducting planning on high-level graph representations and deploying a low-level controller to locally move between nodes; our framework also falls under this paradigm of graph-based planning. Works such as SPTM [29] and SoRB [9] used a deep network as a distance metric for finding shortest paths on the graph, but rely on human demonstrations or sampling from the replay buffer to populate graph nodes. SGM [16] introduced a two-way consistency check to promote sparsity in the graph, allowing these methods to scale to larger maps. However, these methods rely on assumptions about exploration, allowing the agent to spawn uniformly random across the state space during training. To address this, HTM [19] used a generative model to hallucinate samples for building the graph in a zero-shot manner, but was not evaluated in 3D visual environments. NTS [4] achieved exploration and long-horizon navigation as well as generalization to unseen maps by learning a geometric-based graph representation, but required access to a ground-truth map to train their supervised learning model. In contrast, our method achieves exploration by planning and executing paths towards landmarks near novel areas during training. SFL also does not require any ground-truth data; it only needs to learn SF in a self-supervised manner.

**Successor features.** Our work is inspired by recent efforts in developing successor features (SF) [15, 2]. They have used SF to decompose the Q-value function into SF and the reward which enables efficient policy transfer [2, 3] and to design transition-based intrinsic rewards [20, 39] for efficient exploration. SF has also been used in the options framework [35]; Eigenoptions [21] derived options from the eigendecomposition of SF, but did not yet apply them towards reward maximization. Successor Options [28] used SF to discover landmark states and design a latent reward for learning option policies, but was limited to low-dimensional state spaces. Our framework leverages a SF-based similarity metric to formulate a goal-conditioned policy, abstract the state space as a landmark graph for long-horizon planning and to model state-novelty for driving exploration. While the options policies proposed in these works have to be learned from a reward signal, we can obtain our goal-conditioned policy directly from the SF similarity metric without any policy learning. To our knowledge, this is first work that uses SF for graph-based planning and long-horizon GCRL tasks.

## 3 Preliminaries

### 3.1 Goal-Conditioned RL

Goal-conditioned reinforcement learning (GCRL) tasks [12] are Markov Decision Processes (MDP) extended with a set of goals $\mathcal{G}$ and defined by a tuple $(\mathcal{S}, \mathcal{A}, \mathcal{G}, \mathcal{R}_\mathcal{G}, \mathcal{T}, \gamma)$, where $\mathcal{S}$ is a state space, $\mathcal{A}$ an action set, $\mathcal{R}_\mathcal{G} : \mathcal{S} \times \mathcal{A} \times \mathcal{G} \to \mathbb{R}$ a goal-conditioned reward function, $\mathcal{T}$ the transition dynamics, and $\gamma$ a discount factor. Following [11, 36], we focus on the setting where the goal space $\mathcal{G}$ is a subset of the state space $\mathcal{S}$, and the agent can receive non-trivial rewards only when it can reach the goal (*i.e.*, sparse-reward setting). We aim to find an optimal goal-conditioned policy $\pi : \mathcal{S} \times \mathcal{G} \to \mathcal{A}$ to maximize the expected cumulative reward, $V_g^\pi(s_0) = \mathbb{E}^\pi[\sum_t \gamma^t r_t]$; *i.e.*, goal-conditioned value function. We are especially interested in long-horizon tasks where goals are distant from the agent's starting state, requiring the policy to operate over longer temporal sequences.

---

**Algorithm 1** Training

---

1: **Initialize:** Graph $G = (L, E)$, parameter $\theta$ of $\text{SFS}_\theta^{\bar{\pi}}$, replay buffer $D$, hyperparameter $T_{\text{exp}}$, landmark transition count $N^l$
2: **while** env not done **do**
3:    $l_{\text{front}} \sim \text{Softmax}(\frac{1}{\text{Count}(L)})$              {Choose frontier landmark via count-based sampling}
4:    **while** $l_{\text{curr}} \neq l_{\text{front}}$ **do**
5:       $l_{\text{target}} \leftarrow \text{PLAN}(G, l_{\text{curr}}, l_{\text{front}})$                     {Plan path to frontier landmark}
6:       $\tau_{\text{traverse}}, l_{\text{curr}} \leftarrow \text{Traverse}(\pi_l, l_{\text{target}})$     {Traverse to $l_{\text{target}}$ with $\pi_l$ (Algorithm 3 in §H)}
7:       $G, N^l \leftarrow \text{Graph-Update}(G, \tau_{\text{traverse}}, \text{SFS}_\theta^{\bar{\pi}}, N^l)$             {Update graph (Algorithm 2)}
8:    **end while**
9:    $\tau_{\text{rand}} = \{s_t, a_t, r_t\}_t^{T_{\text{exp}}} \sim \bar{\pi}$                {Explore with random policy for $T_{\text{exp}}$ steps}
10:   $G, N^l \leftarrow \text{Graph-Update}(G, \tau_{\text{rand}}, \text{SFS}_\theta^{\bar{\pi}}, N^l)$             {Update graph (Algorithm 2)}
11:   $D \leftarrow D \cup \tau_{\text{random}}$
12:   Update $\theta$ from TD error with mini-batches sampled from $D$       {Update SF parameters}
13: **end while**

---

## 3.2 Successor Features

In the tabular setting, the successor representation (SR) [8, 15] is defined as the expected discounted occupancy of futures state $s'$ starting in state $s$ and action $a$ and acting under a policy $\pi$:

$$M_\pi(s, a, s') = \mathbb{E}^\pi \left[ \sum_{t'=t}^\infty \gamma^{t'-t} \mathbb{I}(S_{t'} = s') \Big| S_t = s, A_t = a \right] \tag{1}$$

The SR $M(s, a)$ is then a concatenation of $M(s, a, s'), \forall s \in S$. We may view SR as a representation of state similarity extended over the time dimension, as described in [21]. In addition, we note that the SR is solely determined by $\pi$ and the transition dynamics of the environment $p(s_{t+1}|s_t, a_t)$.

Successor features (SF) [2, 15] extend SR [8] to high-dimensional, continuous state spaces in which function approximation is often used. SF's formulation modifies the definition of SR by replacing enumeration over all states $s'$ with feature vector $\phi_{s'}$. The SF $\psi^\pi$ of a state-action pair $(s, a)$ is then defined as:

$$\psi^\pi(s, a) = \mathbb{E}^\pi \left[ \sum_{t'=t}^\infty \gamma^{t'-t} \phi_{s_{t'}} \Big| S_t = s, A_t = a \right] \tag{2}$$

In addition, SF can be defined in terms of only the state, $\psi^\pi(s) = \mathbb{E}_{a \sim \pi(s)}[\psi^\pi(s, a)]$.

SF allows decoupling the value function into the successor feature (dynamics-relevant information) with task (reward function): if we assume that the one-step reward of transition $(s, a, s')$ with feature $\phi(s, a, s')$ can be written as $r(s, a, s') = \phi(s, a, s')^\top \mathbf{w}$, where $\mathbf{w}$ are learnable weights to fit the reward function, we can write the Q-value function as follows [2, 15]:

$$Q^\pi(s, a) = \mathbb{E}^\pi \left[ \sum_{t'=t}^\infty \gamma^{t'-t} r(S_{t'}, A_{t'}, S_{t'+1}) \Big| S_t = s, A_t = a \right]$$

$$= \mathbb{E}^\pi \left[ \sum_{t'=t}^\infty \gamma^{t'-t} \phi_{t'}^\top \mathbf{w} \Big| S_t = s, A_t = a \right] = \psi^\pi(s, a)^\top \mathbf{w} \tag{3}$$

Consequently, the Q-value function separates into SF, which represents the policy-dependent transition dynamics, and the reward vector $\mathbf{w}$ for a particular task. Later in Section 4.3, we will extend this formulation to the goal-conditioned setting and discuss our choices for $\mathbf{w}$.

## 4 Successor Feature Landmarks

We present Successor Feature Landmarks (SFL), a graph-based planning framework for supporting exploration and long-horizon GCRL. SFL is centrally built upon our novel distance metric: Successor Feature Similarity (SFS, §4.1). We maintain a non-parametric graph of state "landmarks," using SFS as a distance metric for determining which observed states to add as landmarks and how these landmarks should be connected (§4.2). To enable traversal between landmarks, we directly obtain a local goal-conditioned policy from SFS between current state and the given landmark (§4.3). With these components, we tackle the long-horizon setting by planning on the landmark graph and finding

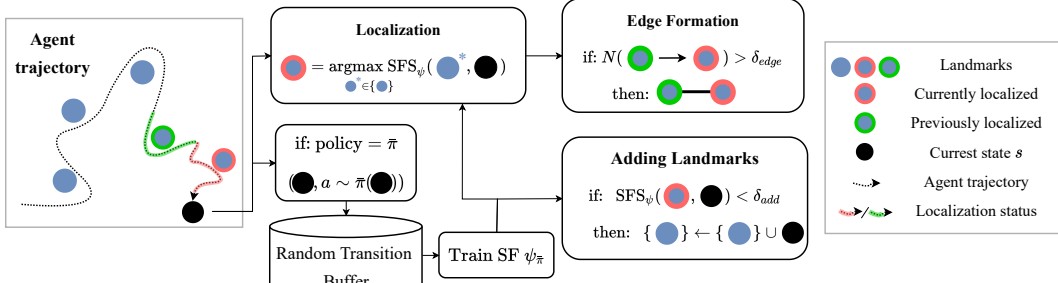

Figure 2: The **Graph + SF Update** step occurs after every transition $(s, a)$. The agent computes SFS between current state $s$ (black dot) and all landmarks (blue dots). It then localizes itself to the nearest landmark (red circled dot). The agent records transitions between the previously localized landmark (green circled dot) and this landmark. If the number of transitions between two landmarks is greater than $\delta_{edge}$, then an edge is formed between them. If SFS between the current localized landmark and $s$ is less than $\delta_{add}$, then $s$ is added as a landmark. Finally, transitions generated by the random policy are added to the random transition buffer, and SF is trained on batch samples from this buffer.

the shortest path to the given goal, which decomposes the long-horizon problem into a sequence of short-horizon tasks that the local policy can then more reliably achieve (§4.4). In training, our agent focuses on exploration. We set the goals as "frontier" landmarks lying at the edge of the explored region, and use the planner and local policy to reach the frontier landmark. Upon reaching the frontier, the agent locally explores with a random policy and uses this new data to update its SF and landmark graph (§4.5). In evaluation, we add the given goal to the graph and follow the shortest path to it. Figure 1 illustrates the overarching framework and Figure 2 gives further detail into how the graph and SF are updated. Algorithm 1 describes the procedure used to train SFL.

## 4.1 Successor Feature Similarity

SFS, the foundation of our framework, is based on SF. For context, we estimate SF $\psi$ as the output of a deep neural network parameterized by $\theta : \psi^{\pi}(s, a) \approx \psi_{\theta}^{\pi}(\phi(s), a)$, where $\phi$ is a feature embedding of state and $\pi$ is a fixed policy which we choose to be a uniform random policy denoted as $\bar{\pi}$. We update $\theta$ by minimizing the temporal difference (TD) error [15, 2]. Details on learning SF are provided in Appendix C.3.

Next, to gain intuition for SFS, suppose we wish to compare two state-action pairs $(s_1, a_1)$ and $(s_2, a_2)$ in terms of similarity. One option is to compare $s_1$ and $s_2$ directly via some metric such as $\ell_2$ distance, but this ignores $a_1, a_2$ and dynamics of the environment.

To address this issue, we should also consider the states the agent is expected to visit when starting from each state-action pair, for a fixed policy $\pi$. We choose $\pi$ to be uniform random, i.e. $\bar{\pi}$, so that only the dynamics of the environment will dictate which states the agent will visit. With this idea, we can define a novel similarity metric, **Successor Feature Similarity (SFS)**, which measures the similarity of the expected discounted state-occupancy of two state-action pairs. Using the successor representation $M_{\bar{\pi}}(s, a)$ as defined in Section 3.2, we can simply define SFS as the dot-product between the two successor representations for each state-action pair:

$$
\begin{aligned}
&\text{SFS}^{\bar{\pi}}\big((s_1, a_1), (s_2, a_2)\big) \\
&= \sum_{s' \in S} \mathbb{E}^{\bar{\pi}}\left[\sum_{t'=t}^{\infty} \gamma^{t'-t} \mathbb{I}(S_t = s') \,\middle|\, \begin{matrix} S_t = s_1, \\ A_t = a_1 \end{matrix}\right] \times \mathbb{E}^{\bar{\pi}}\left[\sum_{t'=t}^{\infty} \gamma^{t'-t} \mathbb{I}(S_t = s') \,\middle|\, \begin{matrix} S_t = s_2, \\ A_t = a_2 \end{matrix}\right] \quad (4) \\
&= \sum_{s' \in S} M_{\bar{\pi}}(s_1, a_1, s') \times M_{\bar{\pi}}(s_2, a_2, s') = M_{\bar{\pi}}(s_1, a_1)^{\top} M_{\bar{\pi}}(s_2, a_2)
\end{aligned}
$$

We can extend SFS to the high-dimensional case by encoding states in the feature space $\phi$ and replacing $M_{\bar{\pi}}(s, a)$ with $\psi^{\bar{\pi}}(s, a)$. The intuition remains the same, but we instead measure similarities in the feature space. In practice, we normalize $\psi^{\pi}(s, a)$ before computing SFS to prevent high-value feature dimensions from dominating the similarity metric, hence defining SFS as the cosine similarity between SF. In addition, we may define SFS between just two states by getting rid of the action

dimension in SF:

$$\text{SFS}^{\bar{\pi}}((s_1, a_1), (s_2, a_2)) = \psi^{\bar{\pi}}(s_1, a_1)^{\top} \psi^{\bar{\pi}}(s_2, a_2) \tag{5}$$

$$\text{SFS}^{\bar{\pi}}(s_1, s_2) = \psi^{\bar{\pi}}(s_1)^{\top} \psi^{\bar{\pi}}(s_2) \tag{6}$$

## 4.2 Landmark Graph

The landmark graph $G$ serves as a compact representation of the state space and its transition dynamics. $G$ is dynamically-populated in an online fashion as the agent explores more of its environment. Formally, landmark graph $G = (L, E)$ is a tuple of landmarks $L$ and edges $E$. The landmark set $L = \{l_1, \ldots, l_{|L|}\}$ is a set of states representing the explored part of the state-space. The edge set $E$ is a matrix $\mathbb{R}^{|L| \times |L|}$, where $E_{i,j} = 1$ if $l_i$ and $l_j$ is connected, and 0 otherwise. Algorithm 2 outlines the graph update process, and the following paragraph describes this process in further detail.

**Agent Localization and Adding Landmarks**   At every time step $t$, we compute the landmark closest to the agent under SFS metric: $l_{\text{curr}} = \arg\max_{l \in L} \text{SFS}^{\bar{\pi}}(s_t, l)$. If $\text{SFS}^{\bar{\pi}}(s_t, l_{\text{curr}}) < \delta_{add}$, the add threshold, then we add $s_t$ to the landmark set $L$. Otherwise, if $\text{SFS}^{\bar{\pi}}(s_t, l_{\text{curr}}) > \delta_{\text{local}}$, the localization threshold, then the agent is localized to $l_{\text{curr}}$. If the agent was previously localized to a different landmark $l_{\text{prev}}$, then we increment the count of the landmark transition $N^l_{(l_{\text{prev}} \to l_{\text{curr}})}$ by 1 where $N^l \in \mathbb{N}^{|E|}$ is the landmark transition count, which is used to form the graph edges. [2]

Since we progressively build the landmark set, we maintain all previously added landmarks. As described above, this enables us to utilize useful landmark metrics such as how many times the agent has been localized to each landmark and what transitions have occurred between landmarks to improve the connectivity quality of the graph. In comparison, landmarks identified through clustering schemes such as in Successor Options [28] cannot be used in this manner because the landmark set is rebuilt every few iterations. See Appendix B.3 for a detailed comparison on landmark formation.

**Edge Formation**   We form edge $E_{i,j}$ if the number of the landmark transitions is larger than the edge threshold, *i.e.*, $N^l_{l_i \to l_j} > \delta_{edge}$, with weight $W_{i,j} = \exp(-(N^l_{l_i \to l_j}))$. We apply filtering improvements to $E$ in **ViZ-Doom** to mitigate the perceptual aliasing problem where faraway states can appear visually similar due to repeated use of textures. See Appendix D for more details.

## 4.3 Local Goal-Conditioned Policy

We want to learn a local goal-conditioned policy $\pi_l : \mathcal{S} \times \mathcal{G} \to \mathcal{A}$ to reach or transition between landmarks. To accomplish this, $\pi_l$ should maximize the expected return $V(s, g) = \mathbb{E}[\sum_{t=0}^{\infty} \gamma^t r(s_t, a_t, g)]$, where $r(s, a, g)$ is a reward function that captures how close $s$ (or more precisely $s, a$) is to the goal $g$ in terms of feature similarity:

$$r(s, a, g) = \phi(s, a)^{\top} \psi^{\bar{\pi}}(g), \tag{7}$$

where $\psi^{\bar{\pi}}$ is the SF with respect to the random policy $\bar{\pi}$. Recall that we can decouple the Q-value function into the SF representation and reward weights $\mathbf{w}$ as shown in

---

**Algorithm 2** Graph-Update (§4.2)

---

**input** Graph $G = (L, E)$, $\text{SFS}^{\bar{\pi}}_{\theta}$, trajectory $\tau$,
     landmark transition count $N^l$
**output** updated graph $G$ and $N^l$
1: $l_{\text{prev}} \leftarrow \emptyset$        {Previously localized landmark}
2: **for** $s \in \tau$ **do**
3:     $l_{\text{curr}} \leftarrow \arg\max_{l \in L} \text{SFS}^{\bar{\pi}}_{\theta}(s, l)$     {Localize}
4:     **if** $\text{SFS}^{\bar{\pi}}_{\theta}(s, l_{\text{curr}}) < \delta_{add}$ **then**
5:        $L \leftarrow L \cup s$        {Add landmark}
6:     **end if**
7:     **if** $\text{SFS}^{\bar{\pi}}_{\theta}(s, l_{\text{curr}}) > \delta_{local}$ **then**
8:        **if** $l_{\text{prev}} \neq \emptyset$ and $l_{\text{prev}} \neq l_{\text{curr}}$ **then**
9:           $N^l_{(l_{\text{prev}} \to l_{\text{curr}})} \leftarrow N^l_{(l_{\text{prev}} \to l_{\text{curr}})} + 1$ {Record landmark transition}
10:          **if** $N^l_{(l_{\text{prev}} \to l_{\text{curr}})} > \delta_{edge}$ **then**
11:             $E \leftarrow E \cup (l_{\text{prev}} \to l_{\text{curr}})$ {Form edge}
12:          **end if**
13:        **end if**
14:        $l_{\text{prev}} \leftarrow l_{\text{curr}}$
15:     **end if**
16: **end for**
17: **return** $G$, $N^l$

---

Eq. (3). The reward function Eq. (7) is our deliberate choice rather than learning a linear reward regression model [2], so the value function can be instantly computed. If we let $\mathbf{w}$ be $\psi^{\bar{\pi}}(g)$, we can

---

[2]Zhang et al. [38] proposed a similar idea of recording the transitions between sets of user-defined attributes. We extend this idea to the function approximation setting where landmark attributes are their SFs.

have the Q-value function $Q^{\bar\pi}(s, a, g)$ for the goal-conditioned policy $\pi(a|s, g)$ being equal to the SFS between $s$ and $g$:

$$Q^{\bar\pi}(s, a, g) = \psi^{\bar\pi}(s, a)^\top \psi^{\bar\pi}(g) = \text{SFS}^{\bar\pi}(s, a, g). \tag{8}$$

The goal-conditioned policy is derived by sampling actions from the goal-conditioned Q-value function in a greedy manner for discrete actions. In the continuous action case, we can learn the goal-conditioned policy by using a compatible algorithm such as DDPG [18]. However, extending SF learning to continuous action spaces is beyond the scope of this work and is left for future work.

### 4.4 Planning

Given the landmark graph $G$, we can plan the shortest-path from the landmark closest to the agent $l_{\text{curr}}$ to a final target landmark $l_{\text{target}}$ by selecting a sequence of landmarks $[l_0, l_1, \ldots, l_k]$ in the graph $G$ with minimal weight (see §4.2) sum along the path, where $l_0 = l_{\text{curr}}$, $l_k = l_{\text{target}}$, and $k$ is the length of the plan. In training, we use frontier landmarks $l_{\text{front}}$ which have been visited less frequently as $l_{\text{target}}$. In evaluation, the given goal state is added to the graph and set as $l_{\text{target}}$. See Algorithm 1 for an overview of how planning is used to select the agent's low-level policy in training.

### 4.5 Exploration

We sample frontier landmarks $l_{\text{front}}$ proportional to the inverse of their visitation count (*i.e.*, with count-based exploration). We use two policies: a local policy $\pi_l$ for traversing between landmarks and a random policy $\bar\pi$ for exploring around a frontier landmark. Given a target frontier landmark $l_{\text{front}}$, we construct a plan $[l_0, l_1, \ldots, l_{\text{front}}]$. When the agent is localized to landmark $l_i$, the policy at time $t$ is defined as $\pi_l(a|s_t; l_{i+1})$. When the agent is localized to a landmark that is not included in the current plan $[l_0, l_1, \ldots, l_{\text{front}}]$, then it re-plans a new path to $l_{\text{front}}$. Such failure cases of transition between the landmarks are used to prevent edge between those landmarks from being formed (see Appendix D for details). We run this process until either $l_{\text{front}}$ is reached or until the step-limit $N_{\text{front}}$ is reached. At that point, random policy $\bar\pi$ is deployed for exploration for $N_{\text{explore}}$ steps, adding novel states to our graph as a new landmark. While the random policy is deployed, its trajectory $\tau \sim \bar\pi$ is added to the random transition buffer. SF $\psi_\theta^{\bar\pi}$ is updated with batch samples from this buffer.

The random policy is only used to explore local neighborhoods at frontier regions for a short horizon while the goal-conditioned policy and planner are responsible for traveling to these frontier regions. Under this framework, the agent is able to visit a diverse set of states in a relatively efficient manner. Our experiments on large **ViZDoom** maps demonstrate that this strategy is sufficient for learning a SF representation that ultimately outperforms baseline methods on goal-reaching tasks.

## 5 Experiments

In our experiments, we evaluate the benefits of SFL for exploration and long-horizon GCRL. We first study how well our framework supports reaching long-horizon goals when the agent's start state is randomized across episodes. Afterwards, we consider how SFL performs when efficient exploration is required to reach distant areas by using a setting where the agent's start state is fixed in training.

### 5.1 Domain and Goal-Specification

**ViZDoom** is a visual navigation environment with 3D first-person observations. In **ViZDoom**, the large-scale of the maps and reuse of similar textures make it particularly difficult to learn distance metrics from the first-person observations. This is due to perceptual aliasing, where visually similar images can be geographically far apart. We use mazes from SPTM in our experiments, with one example shown in Figure 3 [29].

**MiniGrid** is a 2D gridworld with tasks that require the agent to overcome different obstacles to access portions of the map. We experiment on **FourRoom**, a basic 4-room map, and **MultiRoom**, where the agent needs to open doors to reach new rooms.

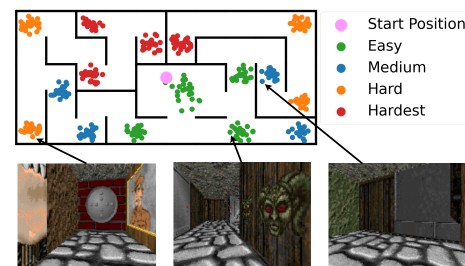

Figure 3: Top-down view of a **ViZDoom** maze used in *fixed spawn* with sampled goal locations. Examples of image goals given to the agent are shown at the bottom. Note that the agent cannot access the top-down view.

We study two settings:

1. **Random spawn**: In training, the agent is randomly spawned across the map with no given goal. In evaluation, the agent is then tested on pairs of start and goal states, where goals are given as images. This enables us to study how well the landmark graph supports traversal between arbitrary start-goal pairs. Following [16], we evaluate on `easy`, `medium`, and `hard` tasks where the goal is sampled within 200m, 200-400m, and 400-600m from the initial state, respectively.

2. **Fixed spawn**: In training, the agent is spawned at a fixed start state $s_{\text{start}}$ with no given goal. This enables us to study how well the agent can explore the map given a limited step budget per episode. In evaluation, the agent is again spawned at $s_{\text{start}}$ and is given different goal states to reach. In **ViZDoom**, we sample goals of varying difficulty accounting for the maze structure as shown in Figure 3. In **MiniGrid**, a similar setup is used except only one goal state is used for evaluation.

## 5.2 Baseline Methods for Comparison

**Random spawn experiments**. We compare SFL against baselines used in SGM, as described below. For each difficulty, we measure the average success rate over 5 random seeds. We evaluate on the map used in SGM, *SGM-Map*, and two more maps from SPTM, *Test-2* and *Test-6*.

1. **Random Actions**: random acting agent. Baseline shows task difficulty.
2. **Visual Controller**: model-free visual controller learned via inverse dynamics. Baseline highlights how low-level controllers struggle to learn long-horizon policies and the benefits of planning to create high-level paths that the controller can follow.
3. **SPTM** [29]: planning module with a reachability network to learn a distance metric for localization and landmark graph formation. Baseline is used to measure how the SFS metric can improve localization and landmark graph formation.
4. **SGM** [16]: data structure used to improve planning by inducing sparsity in landmark graphs. Baseline represents a recent landmark-based approach for long-horizon navigation.

**Fixed spawn experiments**. In the *fixed spawn* setting, we compare SFL against Mapping State Space (MSS) [11], a UVFA and landmark-based approach for exploration and long-horizon goal-conditioned RL, as well as SPTM and SGM. Again, we measure the average success rate over 5 random seeds. We adapt the published code[3] to work on **ViZDoom** and **MiniGrid**. To evaluate SPTM and SGM, we populate their graphs with exploration trajectories generated by Episodic Curiosity (EC) [30]. EC learns an exploration policy by using a reachability network to determine whether an observation is novel enough to be added to the memory and rewarding the agent every time one is added. Appendix C further discusses the implementation of these baselines.

## 5.3 Implementation Details

SFL is implemented with the *rlpyt* codebase [33]. For experiments in **ViZDoom**, we use the pretrained ResNet-18 backbone from SPTM as a fixed feature encoder which is similarly used across all baselines. For **MiniGrid**, we train a convolutional feature encoder using time-contrastive metric learning [32]. Both feature encoders are trained in a self-supervised manner and aim to encode temporally close states as similar feature representations and temporally far states as dissimilar representations. We then approximate SF with a fully-connected neural network, using these encoded features as input. See Appendix C for more details on feature learning, edge formation, and hyperparameters.

## 5.4 Results

**Random Spawn Results**. As shown in Table 1, our method outperforms the other baselines on all settings. SFL's performance on the Hard setting particularly illustrates its ability to reach long-horizon goals. In terms of sample efficiency, SFL utilizes a total of 2M environment steps to simultaneously train SF and build the landmark graph. For reference, SPTM and SGM train their reachability and low-level controller networks with over 250M environment steps of training data collected on *SGM-Map*, with SGM using an additional 114K steps to build and cleanup their landmark graph. For *Test-2* and *Test-6*, we fine-tune these two networks with 4M steps of training data collected from each new map to give a fair comparison.

---

[3] https://github.com/FangchenLiu/map_planner

| Method | SGM-Map | | | Test-2 | | | Test-6 | | |
|---|---|---|---|---|---|---|---|---|---|
| | Easy | Medium | Hard | Easy | Medium | Hard | Easy | Medium | Hard |
| Random Actions | 58% | 22% | 12% | 70% | 39% | 16% | 80% | 31% | 18% |
| Visual Controller | 75% | 35% | 19% | 83% | 51% | 30% | 89% | 39% | 20% |
| SPTM [29] | 70% | 34% | 14% | 78% | 48% | 18% | 88% | 40% | 18% |
| SGM [16] | **92%** | 64% | 26% | **86%** | 54% | 32% | 83% | 43% | 27% |
| SFL [Ours] | **92%** | **82%** | **67%** | 82% | **66%** | **48%** | **92%** | **66%** | **60%** |

Table 1: (*Random spawn*) The success rates of compared methods on three **ViZDoom** maps.

| Method | Test-1 | | | | Test-4 | | | |
|---|---|---|---|---|---|---|---|---|
| | Easy | Medium | Hard | Hardest | Easy | Medium | Hard | Hardest |
| MSS [11] | 23% | 9% | 1% | 1% | 21% | 7% | 7% | 7% |
| EC [30] + SPTM [29] | 48% | 16% | 2% | 0% | 20% | 10% | 4% | 0% |
| EC [30] + SGM [16] | 43% | 3% | 0% | 0% | 28% | 7% | 4% | 1% |
| SFL [Ours] | **85%** | **59%** | **62%** | **50%** | **66%** | **44%** | **27%** | **23%** |

Table 2: (*Fixed spawn*) The success rates of compared methods on three **ViZDoom** maps.

**Fixed Spawn Results.** We see in Table 2 that SFL reaches significantly higher success rates than the baselines across all difficulty levels, especially on *Hard* and *Hardest*. Figure 4 shows the average success rate over the number of environment steps for SFL (red) and MSS (green). We hypothesize that MSS struggles because its UVFA is unable to capture geodesic distance in **ViZDoom**'s high-dimensional state space with first-person views. The UVFA in MSS has to solve the difficult task of approximating the number of steps between two states, which we conjecture requires a larger sample complexity and more learning capacity. In contrast, we only use SFS to relatively compare states, i.e. is SFS of state A higher than SFS of state B with respect to reference state C? EC-augmented SPTM and SGM partially outperform MSS, but cannot scale to harder difficulty levels. We suggest that these baselines suffer from disjointedness of exploration and planning: the EC exploration module is less effective because it does not utilize planning to efficient reach distant areas, which in turn limits the training of policy networks. See Appendix F and Appendix G for more analysis on the baselines.

Figure 5 shows the average success rate on **MiniGrid** environments, where SFL (red) overall outperforms MSS (green). States in **MiniGrid** encode top-down views of the map with distinct signatures for the agent, walls, and doors, making it easier to learn distance metrics. In spite of this, the environment remains challenging due to the presence of doors as obstacles and the limited time budget per episode.

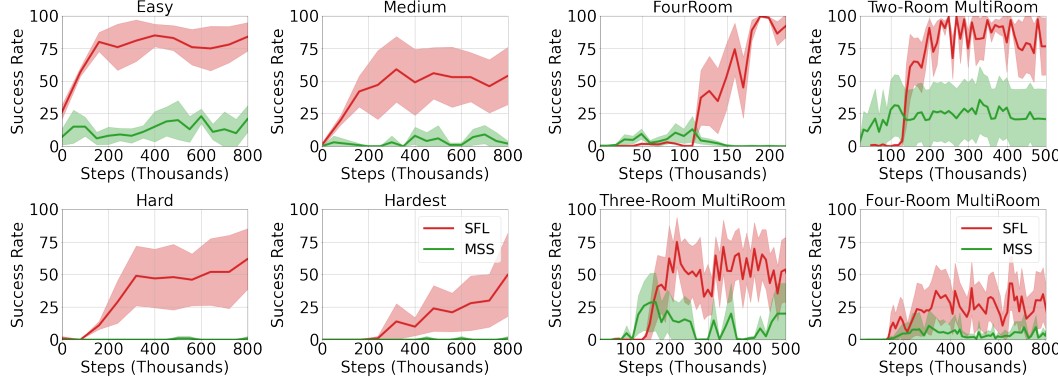

Figure 4: *Fixed spawn* experiments on **ViZDoom** comparing SFL (red) to MSS (green) over number of environment steps for varying difficulty levels.

Figure 5: *Fixed spawn* experiments on **MiniGrid** comparing SFL (red) to MSS (green) over number of environment steps for varying difficulty levels.

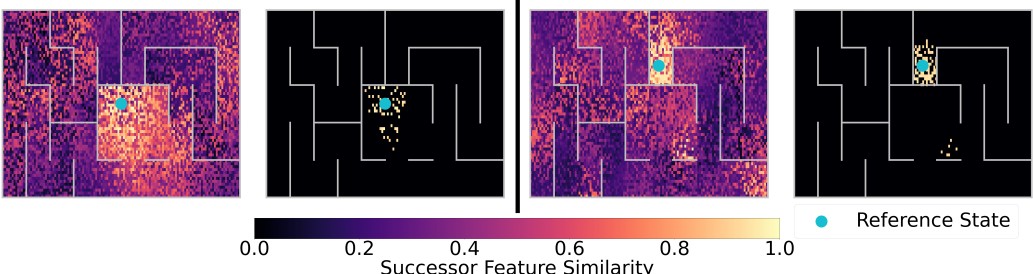

Figure 6: SFS values relative to a reference state (blue dot) in the *Test-1* **ViZDoom** maze. The left two heatmaps use the agent's start state as the reference state while the right two use a distant goal state as the reference state. The first and third (colorful) heatmaps depict all states while the second and fourth (darkened) heatmaps only show states with SFS > $\delta_{\text{local}} = 0.9$.

## 5.5 SFS Visualization

SFL primarily relies on SFS and its capacity to approximate geodesic distance imposed by the map's structure. To provide evidence of this ability, we compute the SFS between a reference state and a set of randomly sampled states. Figure 6 visualizes these SFS heatmaps in a **ViZDoom** maze. In the first and third panels, we observe that states close to the reference state (blue dot) exhibit higher SFS values while distant states, such as those across a wall, exhibit lower SFS values. The second and fourth panels show states in which the agent would be localized to the reference state, i.e. states with SFS > $\delta_{\text{local}}$. With this SFS threshold, we reduce localization errors, thereby improving the quality of the landmark graph. We provide additional analysis of SFS-derived components, the landmark graph and goal-conditioned policy, in Appendix E.

## 6   Conclusion

In this paper, we presented Successor Feature Landmarks, a graph-based planning framework that leverages a SF similarity metric, as an approach to exploration and long-horizon goal-conditioned RL. Our experiments in ViZDoom and MiniGrid, demonstrated that this method outperforms current graph-based approaches on long-horizon goal-reaching tasks. Additionally, we showed that our framework can be used for exploration, enabling discovery and reaching of goals far away from the agent's starting position. Our work empirically showed that SF can be used to make robust decisions about environment dynamics, and we hope that future work will continue along this line by formulating new uses of this representation. Our framework is built upon the representation power of SF, which depends on a good feature embedding to be learned. We foresee that our method can be extended by augmenting with an algorithm for learning robust feature embeddings to facilitate SF learning.

## Acknowledgements

This work was supported by the NSF CAREER IIS 1453651 Grant. JC was partly supported by Korea Foundation for Advanced Studies. WC was supported by an NSF Fellowship under Grant No. DGE1418060. Any opinions, findings, and conclusions or recommendations expressed in this material are those of the author(s) and do not necessarily reflect the views of the funding agencies. We thank Yunseok Jang and Anthony Liu for their valuable feedback. We also thank Scott Emmons and Ajay Jain for sharing and helping with the code for the SGM [16] baseline.

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
