# Supplementary Material
## Successor Feature Landmarks for Long-Horizon Goal-Conditioned Reinforcement Learning

## A  Additional Results

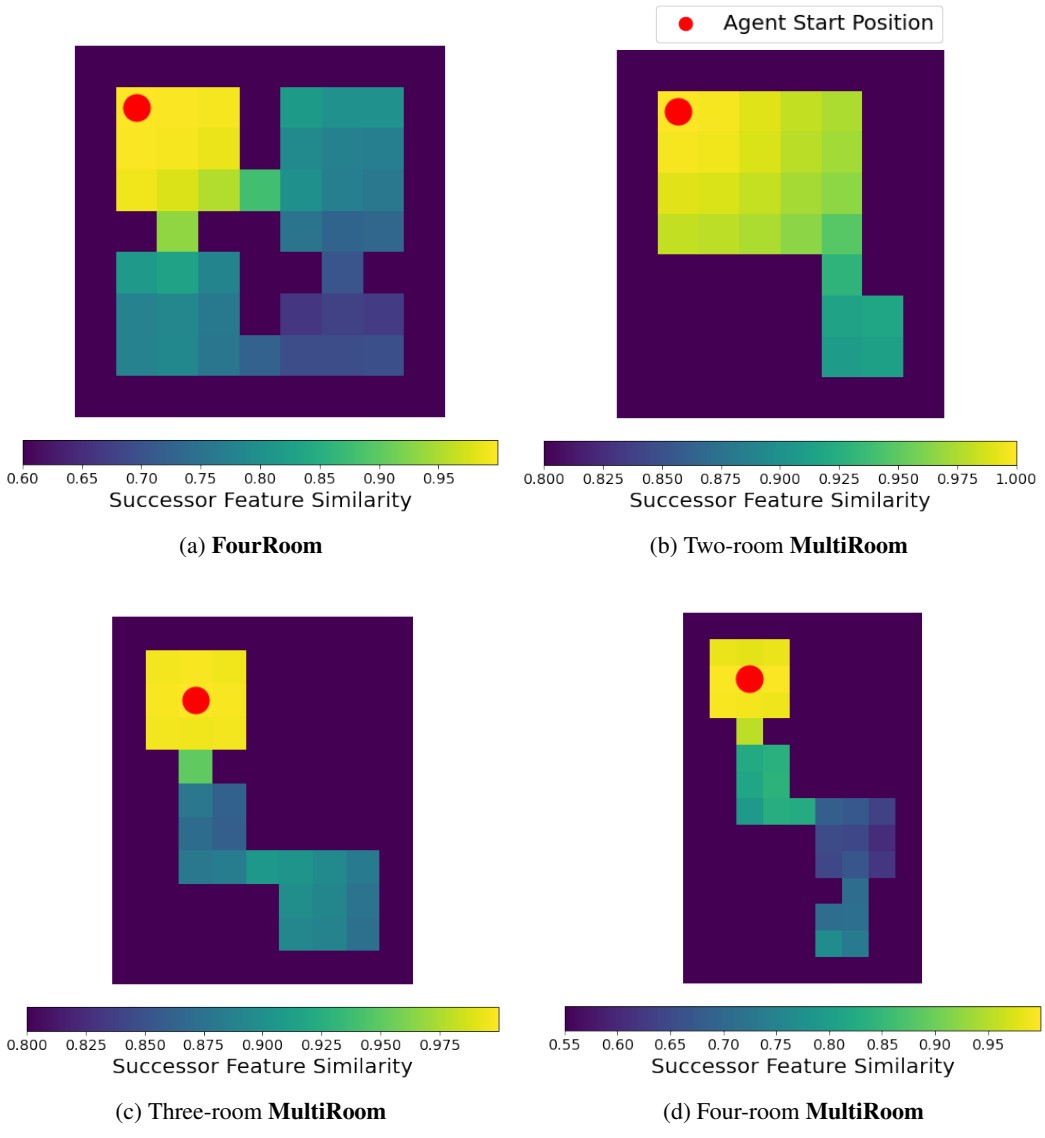

(a) **FourRoom**

(b) Two-room **MultiRoom**

(c) Three-room **MultiRoom**

(d) Four-room **MultiRoom**

Figure 7: SFS values relative to the agent's starting state (red dot) for the different **MiniGrid** environments.

## A.1  MiniGrid

We show visualizations of Successor Feature Similarity (SFS) in the **MiniGrid** environment to further illustrate the metric's capacity to capture distance. Specifically, we compute the SFS between the agent's starting state and the set of possible states and present these values as SFS heatmaps in Figure 7 below. The SFS is distinctly higher for states that reside in the same room as the reference

state (red dot). Additionally, the SFS values gradually decrease as you move further away from the reference state. This effect is most clearly demonstrated in the SFS heatmap of Two-room **MultiRoom** (top right).

## A.2 ViZDoom

We report the standard error for the *random spawn* experiments on **ViZDoom**. The experiments are run over 5 random seeds. Note that we use the reported results from the original SGM paper [16] for the *SGM-Map* and therefore do not report standard errors.

| Method | *SGM-Map* | | | *Test-2* | | | *Test-6* | | |
|---|---|---|---|---|---|---|---|---|---|
| | Easy | Medium | Hard | Easy | Medium | Hard | Easy | Medium | Hard |
| Random Actions | 58% | 22% | 12% | $70 \pm 1.2\%$ | $39 \pm 1.0\%$ | $16 \pm 1.0\%$ | $80 \pm 0.4\%$ | $31 \pm 1.0\%$ | $18 \pm 0.7\%$ |
| Visual Controller | 75% | 35% | 19% | $83 \pm 0.7\%$ | $51 \pm 0.5\%$ | $30 \pm 0.7\%$ | $89 \pm 0.7\%$ | $39 \pm 1.1\%$ | $20 \pm 1.0\%$ |
| SPTM [29] | 70% | 34% | 14% | $78 \pm 0.0\%$ | $48 \pm 0.0\%$ | $18 \pm 0.0\%$ | $88 \pm 0.0\%$ | $40 \pm 0.0\%$ | $18 \pm 0.0\%$ |
| SGM [16] | **92%** | 64% | 26% | $\mathbf{86 \pm 0.8\%}$ | $54 \pm 0.7\%$ | $32 \pm 0.7\%$ | $83 \pm 0.7\%$ | $43 \pm 1.2\%$ | $27 \pm 1.5\%$ |
| SFL [Ours] | $\mathbf{92 \pm 0.8\%}$ | $\mathbf{82 \pm 0.6\%}$ | $\mathbf{67 \pm 1.2\%}$ | $82 \pm 0.7\%$ | $\mathbf{66 \pm 0.8\%}$ | $\mathbf{48 \pm 1.5\%}$ | $\mathbf{92 \pm 0.6\%}$ | $\mathbf{66 \pm 0.7\%}$ | $\mathbf{60 \pm 0.5\%}$ |

Table 3: (*Random spawn*) The success rates and standard errors of compared methods on three **ViZDoom** maps.

# B   Ablation Experiments

We conduct various ablation experiments to isolate and better demonstrate the impact of individual components of our framework.

## B.1   Distance Metric

| Metric | Easy | Medium | Hard |
|---|---|---|---|
| SFS [Ours] | $92 \pm 1.7\%$ | $82 \pm 0.6\%$ | $67 \pm 2.6\%$ |
| SPTM's Reachability Network [29] | $83 \pm 1.2\%$ | $57 \pm 2.6\%$ | $24 \pm 1.9\%$ |

Table 4: The success rates and standard errors of our method and the reachability network ablation on **ViZDoom** *SGM-Map* in the *random spawn* setting.

We compare our SFS metric against the reachability network proposed in SPTM [29] and reused in SGM [16]. In Table 4, we observe that SFS outperforms the reachability network on all difficulty levels, indicating that SFS can more accurately represent transition distance between states than the reachability network. The results are aggregated over 5 random seeds.

## B.2   Exploration Strategy

We investigate the benefit of our exploration strategy, which samples frontier landmarks based on inverse visitation count to travel to before conducting random exploration. We compare against an ablation which samples frontier landmarks from the landmark set in a uniformly random manner, which is analogous to how SGM [16] chooses goals in their cleanup step. We directly measure the degree of exploration achieved by each strategy by tracking state coverage, which we define as the thresholded state visitation count computed over a discretized grid of agent states and report as a percentage over all potentially reachable states. We report the mean state coverage percentage and associated standard error achieved by the two exploration strategies on the *Test-1* **ViZDoom** map over 5 random seeds. Our exploration strategy achieves $79.4 \pm 0.65\%$ state coverage while the uniform random sampling ablation strategy achieves $72.3 \pm 1.90\%$ state coverage, thus indicating that our strategy empirically attains greater exploration of the state space.

### B.3 Landmark Formation

We compare our progressive building of the landmark set to a clustering scheme akin to the one presented in Successor Options [28]. To illustrate the primary benefit of our approach, the ability to track landmark metadata over time, we conduct an experiment with the clustering scheme as an ablation on Three-room **MultiRoom**. Our progressive landmark scheme achieves a mean success rate of $75.0 \pm 14.6\%$ while clustering achieves a rate of $35.6 \pm 18.8\%$, with results aggregated over 5 random seeds. We observe that our method more than doubles the success rate attained by the clustering method and attribute this outperformance to the beneficial landmark information that we are able to record and utilize in constructing the landmark graph.

We also note a secondary benefit in which landmarks are chosen. Our approach aims to minimize the distance between chosen landmarks parameterized by $\delta_{add}$ while clustering selects landmarks which are closer to the center of topologically distinguishable regions. The former method will add landmarks that are far away from existing landmarks, making them more likely to lie on the edge of the explored state space by nature This in turn can improve exploration via our frontier strategy. An experiment on **FourRoom** empirically demonstrates this effect, where the average pairwise geodesic distance between landmarks was $6.72 \pm 0.43$ for our method versus $5.72 \pm 0.40$ for clustering.

## C  Implementation Details

### C.1  Environments and Evaluation Settings

**ViZDoom:** The **ViZDoom** visual environment produces $160 \times 120$ RGB first-person view images as observations. We stacked a history of the 4 most recent observations as our state. We adopted the same action set as SPTM and SGM: *DO NOTHING, MOVE FORWARD, MOVE BACKWARD, MOVE LEFT, MOVE RIGHT, TURN LEFT, TURN RIGHT*. As commonly done for **ViZDoom** in previous works, we used an action repetition of 4. For training, each episode has a time limit of 10,000 steps or 2,500 states after applying action repetition. We reuse the same texture sets as SPTM and SGM for all mazes. Figure 8 shows the maps used in the *random spawn* experiments and Figure 9 shows the maps used in the *fixed spawn* experiments.

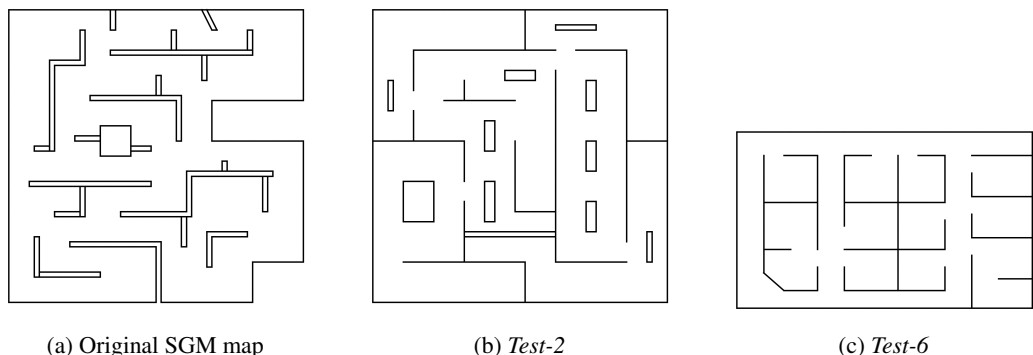

| (a) Original SGM map | (b) *Test-2* | (c) *Test-6* |

Figure 8: **ViZDoom** maps used in *random spawn* experiments.

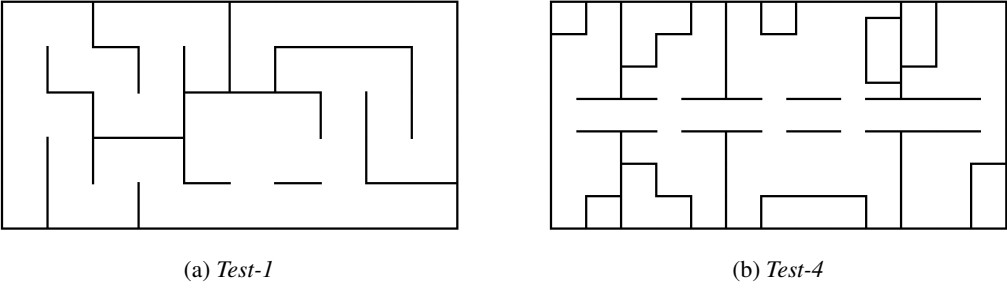

| (a) *Test-1* | (b) *Test-4* |

Figure 9: **ViZDoom** maps used in *fixed spawn* experiments.

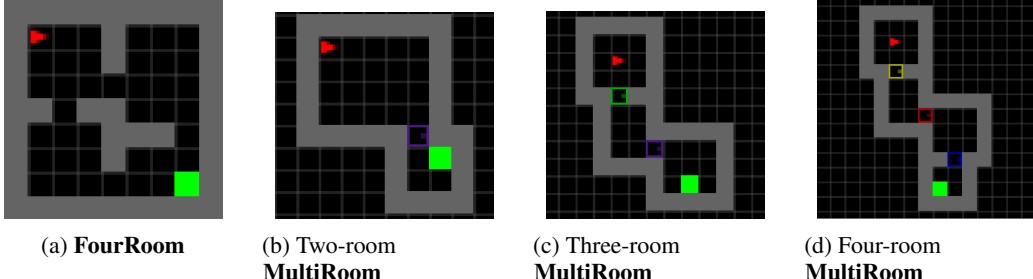

| (a) **FourRoom** | (b) Two-room **MultiRoom** | (c) Three-room **MultiRoom** | (d) Four-room **MultiRoom** |

Figure 10: **MiniGrid** maps used in *fixed spawn*. The agent spawns at red arrow and attempts to reach the goal depicted by the green box.

During evaluation, the goal state is given as a first-person image observation at the goal position, stacked 4 times. For both the *random spawn* and *fixed spawn* settings, the success rate for each difficulty level is computed by evaluating the agent on 50 different start-goal pairs generated by each respective sampling procedure. The reported success rate is the success rate of reaching the goal averaged over 5 random seeds.

**MiniGrid:** The **MiniGrid** environment provides a compact encoding of a top-down view of the maze as the state. The encoding represents each object type such as the agent or doors as a distinct 3-length tuple. The state is $25 \times 25 \times 3$ for **MultiRoom** and $9 \times 9 \times 3$ for **FourRoom**. We use the following action set: *MOVE FORWARD, TURN LEFT, TURN RIGHT, OPEN/CLOSE DOOR*. For training, each episode has a time limit of 100 steps for **FourRoom** and $n_{\text{rooms}} \cdot 40$ steps for $n_{\text{rooms}}$-room **MultiRoom**, where $n_{\text{rooms}}$ is the number of rooms.

During evaluation, the goal state is given as the state if the agent had reached the goal. The time limits to reach the goal are equivalent to the episode time limits during training. We compute the success rate of our method and MSS over 100 trajectories each, averaging over 5 random seeds for **FourRoom** and 15 random seeds for **MultiRoom**.

### C.2 Feature Learning

**ViZDoom:** For our experiments in **ViZDoom**, we adopt a similar feature learning setup as SGM by reusing the pretrained ResNet-18 backbone from SPTM as a fixed feature encoder. The network was originally trained with a self-supervised binary classification task: to determine whether a pair of image observations is temporally close within a time step threshold. The network encodes the image observations as 512-length feature vectors. Recalling that our state is a stack of the 4 most recent observations, we use the encoder to individually embed each of the observations, and then concatenate the 4 intermediate feature vectors into a 2048-length feature vector.

**MiniGrid:** For our experiments in **MiniGrid**, we learn features by training a convolutional feature encoder with time-contrastive metric learning [7, 32]. Specifically, we train an encoder $f$ via a triplet loss on 3-tuples consisting of anchor $o^a$, positive $o^p$, and negative observations $o^n$:

$$||f(o^a) - f(o^p)||_2^2 + m < ||f(o^a) - f(o^n)||_2^2 \tag{9}$$

A margin parameter $m = 2$ is used to encourage separation between the (anchor, positive) and the (anchor, negative) pairs. The 3-tuples are randomly sampled from the replay buffer such that if $o^a$ corresponds to time step $t$, then $o^p$ is uniform randomly sampled from observations from the same episode with time $[t - K_p, t + K_p] = [t - 2, t + 2]$. Similarly, $o^n$ is randomly sampled from the time intervals, $[t - L_n, t - U_n] \cup [t + U_n, t + L_n] = [t - 15, t - 10] \cup [t + 10, t + 15]$.

The encoder network has the following architecture: two $3 \times 3$ convolutional layers with 32 hidden units, and strides 2 and 1 respectively, each followed by ReLU activations, and ending in a linear layer that outputs 64-dimensional feature vectors. Additionally, we normalize the feature vector such that $||\phi(o_t)||_2 = \alpha = 10$ following [32]. The network is trained using the Adam optimizer [14] with a learning rate of $5e - 4$ and a batch size of 128.

## C.3 SF Learning

Recall that we estimate SF $\psi^\pi$ with a deep neural network parameterized by $\theta$ : $\psi^\pi(s, a) \approx \psi_\theta^\pi(\phi(s), a)$. Here, $\phi$ is a feature embedding of state. The parameter $\theta$ is updated via minimizing the temporal difference error [15, 2]:

$$L(s, a, s', \pi, \theta) = \mathbb{E}\left[\left(\phi(s) + \gamma\widehat{\psi}(\phi(s'), \pi(s')) - \psi_\theta^\pi(\phi(s), a)\right)^2\right] \tag{10}$$

where $\widehat{\psi}$ is a target network that is updated at fixed intervals for stability purposes [22]. We choose for $\pi$ to be a fixed uniform random policy because we wish for the SF to capture only the structure and dynamics of the environment and not be biased towards any agent behavior induced by a particular policy. Consequently, we only use transitions from the random policy $\bar{\pi}$ for training $\psi^{\bar{\pi}}$.

We approximate the SF using a fully-connected neural network, which takes in the features (Appendix C.2) as input. Each hidden layer of the network is followed by a batch normalization and ReLU activation layer. For updating the parameters of the SF network, we use Adam to optimize on the TD error loss function shown in Eq. (10) with bootstrapped $n-$step transition tuples. These experience tuples are sampled from a replay buffer of size 100K, which stores trajectories generated by 8 samplers simultaneously running SFL. For stability, we use a target network $\widehat{\psi}$ that is updated at slower intervals and perform gradient clipping as described in [22]. Table 5 describes the hyperparameters used for SF learning in each environment.

| Hyperparameter | ViZDoom | MiniGrid |
|---|---|---|
| Hidden layer units | 2048, 1024 | 512 |
| Learning rate | $1e-4$ | $5e-4$ |
| Batch size | 128 | 128 |
| $n-$step | 5 | 1 |
| Replay buffer size | 100K | 20K |
| Discount $\gamma$ | 0.95 | 0.99 |
| $\widehat{\psi}$ update interval | 1000 | 250 |
| Gradient clip $\delta$ | 5 | 1 |

Table 5: Hyperparameters used in SFL for learning SF for each environment.

| Hyperparameter | ViZDoom | MiniGrid |
|---|---|---|
| $\delta_{add}$ | 0.8 | 0.99 |
| $\delta_{local}$ | 0.9 | 1 |
| $\delta_{edge}$ | - | 1 |
| $N_{front}$ | 1000 | 40 |
| $N_{explore}$ | 500 | 40 |
| $\epsilon_{train}$ | 0.05 | 0.1 |
| $\epsilon_{eval}$ | 0.05 | 0.05 |

Table 6: Hyperparameters used in SFL's landmark graph, planner, and navigation policy for each environment. See Appendix D.2 for details of $\delta_{edge}$ in **ViZDoom**.

Because the cardinality of **MiniGrid**'s state space is much smaller, we restrict SFL to have at most 10 landmarks for **FourRoom** and 30 landmarks for **MultiRoom**, which is consistent with the number of landmarks used in MSS as described in Table 8.

## C.4 SFL Hyperparameters

We mainly tuned these hyperparameters: *learning rate*, $\delta_{add}$, and $\delta_{local}$. In **ViZDoom** for example, we performed grid search over the values of $[10^{-3}, 10^{-4}, 10^{-5}]$ for *learning rate*, $[0.70, 0.75, 0.80, 0.85, 0.90]$ for $\delta_{add}$, and $[0.70, 0.80, 0.9, 0.950]$ for $\delta_{local}$ on the *Train* map.

The best performing values were then used for all other **ViZDoom** experiments. We found that our method performed well under a range of values for $\delta_{add}$, and $\delta_{local}$. In Table 7, we report the success rates achieved on *Hard* tasks from a seed-controlled experiment on the *Train* map for random spawn.

| $\delta_{add}$ | Success Rate |
|---|---|
| 0.70 | 41% |
| 0.80 | 46% |
| 0.90 | 47% |
| $\delta_{local}$ | |
| 0.80 | 24% |
| 0.90 | 46% |
| 0.95 | 44% |
| SGM [16] | 26% |

Table 7: The success rates on *Hard* tasks in *Train* **ViZDoom** map for random spawn for varying values of $\delta_{add}$ and $\delta_{local}$. For reference, SGM is included as the best-performing baseline.

The other hyperparameters were either chosen from related work or not searched over.

## C.5 Optimizations

We perform optimizations on certain parts of SFL for computational efficiency. To add landmarks, we first store states which pass the SFS add threshold, SFS $< \delta_{add}$, to a candidate buffer. Then, $N_{cand}$ landmarks are added from the buffer every $N_{add}$ steps. Additionally, we update the SF representation of the landmarks every $N_{update}$ steps and form edges in the landmark graph every $N_{form-edges}$ steps. Finally, we restrict the step-limit for reaching a frontier landmark to be $n_{land}$ times the number of landmarks on the initially generated path so that we do not overly allocate steps for reaching nearby landmarks.

In **ViZDoom**, $N_{cand} = 50$, $N_{add} = 20K$, $N_{update} = 10K$, $N_{form-edges} = 20K$, $n_{land} = 30$.
In **MiniGrid**, $N_{cand} = 1$, $N_{add} = 3K$, $N_{update} = 1K$, $N_{form-edges} = 1K$, $n_{land} = 8$.

## C.6 Mapping State Space Implementation

We slightly modify the Mapping State Space (MSS) method to work for our environments.

**ViZDoom:** Similar to SGM's and our setup, we reuse the pretrained ResNet-18 network from SPTM as a fixed feature encoder $f$. The UVFA embeds the start and goal states as feature vectors with this encoder, concatenates them into a 4096-length feature vector, and it them through two hidden layers of size 2048 and 1024, each followed by batch normalization and ReLU activation. The outputs are the Q-values of each possible action. The UVFA is trained with HER, which requires a goal-reaching reward function. Because states in **ViZDoom** do not directly give the agent's position, we define the reward function based on the cosine similarity between feature vectors given with $f$:

$$r_t = \mathcal{R}(s_t, a_t, g) = \begin{cases} 0 & f(s_t') \cdot f(g) > \delta_{reach} \\ -1 & \text{otherwise} \end{cases} \tag{11}$$

In the function above, we normalize the feature vectors such that $||f(\cdot)||_2 = 1$ and set $\delta_{reach} = 0.8$.

Landmarks are chosen according to farthest point sampling performed in the feature space imposed by the encoder $f$. During training, the planner randomly chooses a landmark as a goal and attempts to navigate to that goal for 500 steps. The agent then uses an epsilon greedy policy with respect to the Q-values given by the UVFA for a randomly sampled state as the goal for 500 steps. It cycles between these two phases until the episode is over.

| Hyperparameter | ViZDoom | FourRoom | MultiRoom |
|---|---|---|---|
| Learning rate | $10^{-4}$ | $10^{-4}$ | $10^{-3}$ |
| Batch size | 128 | 256 | 128 |
| Replay buffer size | 100K | 100K | 100K |
| Discount $\gamma$ | 0.95 | 0.99 | 0.99 |
| Target update interval | 100 | 50 | 50 |
| Clip threshold | -25 | -3 | -5 |
| Max landmarks | 250 | 10 | 30 |
| HER future step limit | 400 | 10 | 20 |
| Planner $\epsilon_{plan}$ | - | 0.75 | 0.75 |
| Explore $\epsilon_{explore}$ | 0.25 | 0.10 | 0.10 |
| Evaluation $\epsilon_{eval}$ | 0.05 | 0.10 | 0.05 |

Table 8: MSS hyperparameters used for each environment.

**MiniGrid:** The UVFA in **MiniGrid** directly maps observations to action q-values. The UVFA is composed of an encoder with the same architecture as in Appendix C.2 and a fully-connected network with one hidden layer of size 512 followed by a ReLU activation. Like in **ViZDoom**, the UVFA encodes the start and goal states, concatenates the feature vectors together, and passes the output through the fully-connected network. For HER, we reuse the MSS reward function, setting $\delta_{reach} = 1$, i.e. a reward is given only when the agent's next state is the actual goal state. During training, at every time step, the agent will use the planner with epsilon $\epsilon_{plan}$. Otherwise, it will use the epsilon greedy policy like in **ViZDoom** above.

Table 8 describes the hyperparameters we used for each environment, which were determined after rounds of hyperparameter tuning. We give extra attention to the *clip threshold* and *max landmarks* parameters, which MSS [11] mentions are the main hyperparameters of their method.

### C.7 EC-augmented SPTM/SGM Implementation

We use Episodic Curiosity (EC) [30] as an exploration mechanism to enable SPTM and SGM to work in the *fixed spawn* setting on **ViZDoom**. Specifically, we leverage the exploration abilities of EC to generate trajectories that provide greater coverage of the state space. SPTM and SGM then sample from these trajectories to populate their memory graphs and to train their reachability and low-level controller networks. Fortunately, the code repository[4] for EC already has a **ViZDoom** implementation, so minimal changes were required to make it compatible with our experimental setting.

The full procedure is as follows. First, we train EC on the desired evaluation maze and record the trajectories experienced. Second, we take a frozen checkpoint of the EC module and use it to generate trajectories for populating the memory graphs. Third, we train the SPTM/SGM networks on EC's training trajectories recorded in the first step. Last, we run SPTM/SGM as normal with the constructed memory graph and trained networks.

To make the comparison fair with our method and MSS, we load in the weights from the pretrained ResNet-18 reachability network from SPTM as initial weights for EC's reachability network. In the first step, we train EC for 2M environment steps, which is the same number of steps we allow for our method to train. Similar to the *random spawn* experimental setup, we also use the pretrained SPTM/SGM reachability and low-level controller networks, and *fine-tune* them in the third step with 4M steps of training data from the EC-generated trajectories.

We run validation experiments with EC on the original SGM map to search over the following hyperparameters: curiosity bonus scale $\alpha$, EC memory size, and EC novelty threshold. We leverage an oracle exploration bonus based on ground-truth agent coordinates as the validation metric. The same oracle validation metric is used to determine which frozen checkpoint to use in the second step, generating exploration trajectories to populate the memory graph. For the other hyperparameters, we reuse the values chosen for **ViZDoom** in the original EC paper [30]. Table 9 describes the hyperparameters that we selected for EC.

[4]https://github.com/google-research/episodic-curiosity

| Hyperparameter | Value |
| --- | --- |
| Learning rate | $2.5 \times 10^{-4}$ |
| PPO entropy coefficient | 0.01 |
| Curiosity bonus scale $\alpha$ | 0.030 |
| EC memory size | 400 |
| EC reward shift $\beta$ | 0.5 |
| EC novelty threshold $b_{novelty}$ | 0.1 |
| EC aggregation function $F$ | percentile-90 |

Table 9: EC hyperparameters used to generate exploration trajectories for SPTM and SGM.

## C.8 Computational Resources

Each SFL run uses a single GPU and we use 4 GPUs in total to run the SFL experiments: 2 NVIDIA GeForce GTX 1080 and and 2 NVIDIA Tesla K40c. The code was run on a 24-core Intel Xeon CPU @ 2.40 GHz.

## C.9 Asset Licenses

Here is a list of licenses for the assets we use:

1. *rlpyt* code repo: MIT license

2. **ViZDoom** environment: MIT license

3. **MiniGrid** environment: Apache 2.0 license

4. MSS code repo: MIT license

5. SPTM code repo: MIT license

6. SGM code repo: Apache 2.0 license

7. EC code repo: Apache 2.0 license

# D  Tackling Perceptual Aliasing in ViZDoom

Perceptual aliasing is a common problem in visual environments like **ViZDoom**, where two image observations look visually similar, but correspond to distant regions of the environment. This problem can cause our agent to erroneously localize itself to distant landmarks, which in turn harms the accuracy of the landmark graph and planner. Figure 11 gives examples of the perceptual aliasing problem where the pairs of visually similar observations also have very high SFS values relative to each other. We take several steps to make SFL more robust to perceptual aliasing, as described in the following sections.

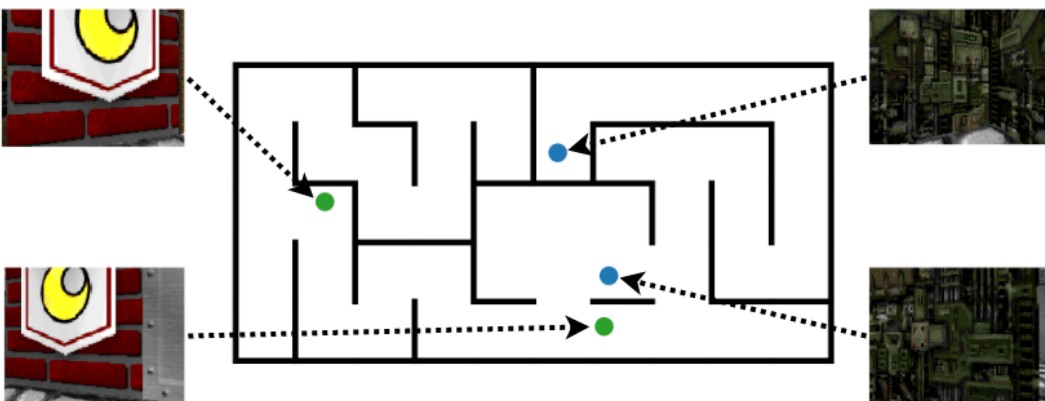

Figure 11: Examples of perceptual aliasing in **ViZDoom**. Same colored dots correspond to pairs of perceptually aliased observations which have pairwise SFS values $> \delta_{local} = 0.9$.

### D.1 Aggregating SFS over Time

We adopt a similar procedure used in SGM and SPTM [16, 29] to make localization more robust by aggregating SFS over a short temporal window. Specifically, we maintain a history of SFS values over the past $W$ steps, and output the median as the final SFS value $S$. This is defined as follows:

$$S = \text{median}([\text{SFS}^{\bar{\pi}}(s_{t-W}, \cdot), ..., \text{SFS}^{\bar{\pi}}(s_t, \cdot)]) \tag{12}$$

where $\text{SFS}^{\bar{\pi}}(s_t, \cdot)$ is the $|L|$-length vector containing $\text{SFS}^{\bar{\pi}}(s_t, l), \forall l \in L$. The median function is taken over the time dimension such that $S$ is length $|L|$. Then, for example, if we were attempting to add $s_t$ as a landmark, we would compute $l_t = \text{argmax}_{l \in L} S_l$ and check if $S_{l_t} < \delta_{add}$ to decide if we should add $s_t$ as a landmark. For our experiments, we set $W = 8$.

### D.2 Edge Threshold

We give special consideration to the edge threshold $\delta_{edge}$ in **ViZDoom** due to the larger scale of the environment and recognized that a fixed edge threshold may not work depending on the stage of training. For example, a low value of $\delta_{edge}$ would allow connections to form in the early stages of training, but would also introduce unwanted noise to the graph as the number of nodes grows. Therefore, we wish for $\delta_{edge}$ to dynamically change depending on the status of the graph. We define it as follows:

$$\delta_{edge} = \text{median}_{l_i \to l_j}(N^l_{l_i \to l_j}) \tag{13}$$

In other words, an edge is formed if the number of landmark transitions on that edge is greater than the median number of landmark transitions from all edges. We found that the median is a suitable threshold for enabling sufficient graph connectivity in the beginning while also reducing the the number of erroneous edges as the graph grows in scale.

### D.3 Edge Filtering

We apply edge filtering to the landmark graph to reduce the number of potential erroneous edges. In *random spawn* experiments, we adopt SGM's $k$-nearest edge filtering where for each vertex, we only keep edges that have the top $k = 5$ number of transitions from that vertex. In *fixed spawn* experiments, we instead introduce *temporal* filtering where we only keep edges between landmarks that were added during similar time periods. Specifically, assuming landmarks are labeled by the order in which they are added, $uv \in G \to |u - v| < \tau_{temporal} \cdot |L|$, where $L$ is the number of landmarks. Our intuition for *temporal* filtering is based on how the agent in the *fixed spawn* setting will add new landmarks further and further away from the starting point as it explores more of the environment over time. Because the agent must pass by the most recently added landmarks at the edge of the already explored areas in order to add new landmarks, landmarks added within similar time periods are overall likely to be closer together. In our experiments, we set $\tau_{temporal} = 0.1$.

Additionally, we adopt the cleanup step proposed in SGM by pruning failure edges. If the agent is unable to traverse an edge with the navigation policy, we remove that edge. To account for the case where the edge is correct, but the goal-conditioned policy has not yet learned how to traverse that edge, we "forget" about failures over time such that only edges that are *repeatedly* untraversable are excluded from the graph. The agent forgets about failures that occurred over 80K steps ago.

These procedures can improve the quality of the landmark graph. Figure 12 shows the landmark graph formed in one of our **ViZDoom** *fixed spawn* experiments when no additional edge filtering steps are used. The graph has many incorrect edges which connect distant landmarks. On the other hand, Figure 13 shows the landmark graph with *temporal* filtering and failure edge cleanup. These procedures eliminate many of the incorrect edges, resulting in a graph which respects the wall structure of the maze to a much higher degree.

## E   SFL Analysis

We conduct further analysis on the components of SFL to study how each contributes to the agent's exploration, goal-reaching, and long-horizon planning abilities.

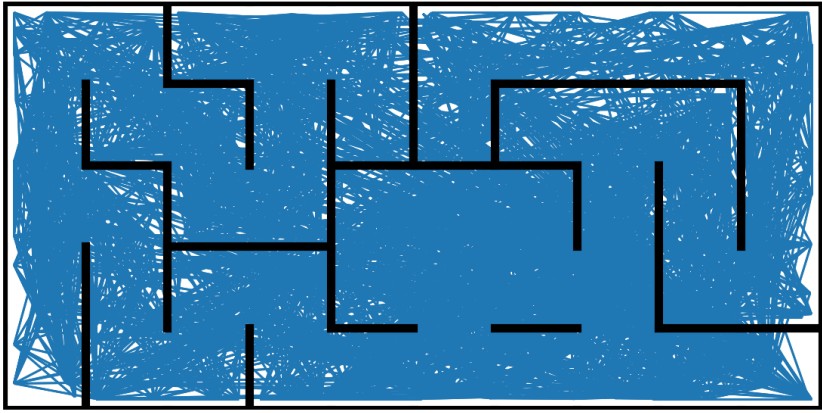

Figure 12: Graph formed with only empirical landmark transitions: $|L| = 517, |E| = 8316$.

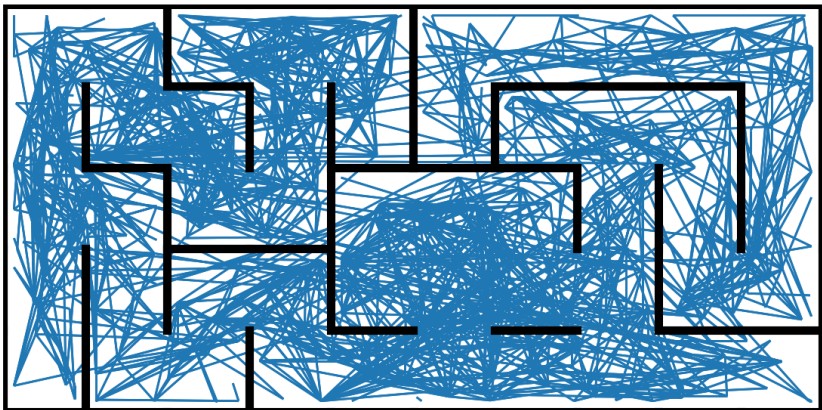

Figure 13: Graph formed with empirical landmark transitions, *temporal filtering*, and failure edge cleanup: $|L| = 517, |E| = 2984$.

### E.1 Exploration

For our *fixed spawn* experiments, as the agent progresses in training, it should spend more time in faraway areas of the state space. In Figure 14a, we show that our agent exhibits this behavior. Each landmark is colored by the relative rank of its visitations, where a lighter color corresponds to spending more time at a landmark. Early in training (top), the agent has discovered some faraway landmarks, but does not spent much time in these distant areas as indicated by the darker color of these landmarks. Later in training (bottom), the agent has both added more landmarks and spent more time near distant landmarks. This is also shown by how the lighter colors are more evenly distributed across the map. We expect agents without effective exploration strategies to remain near the center of the map.

### E.2 Goal-Conditioned Policy

We study how the goal-conditioned policy improves over training. Figure 14b shows how the goal-conditioned policy's success rate over certain landmark edges increases over time, with success

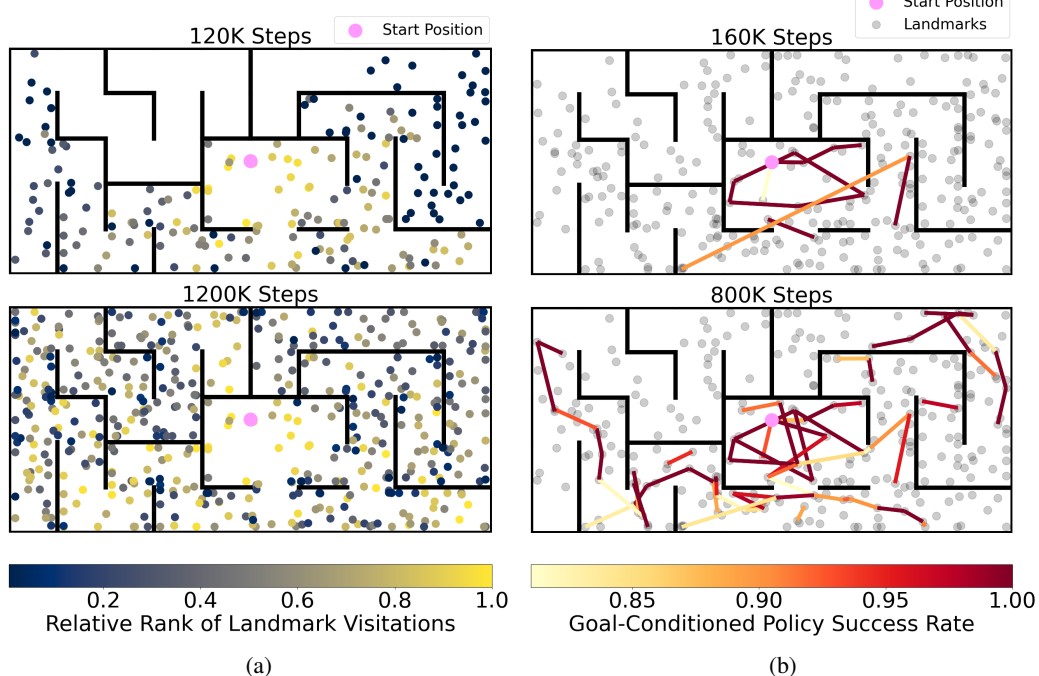

Figure 14: Visualizations of exploration (left) and goal-conditioned policy (right) on the *fixed spawn Test-1* **ViZDoom** maze. **Left:** Each landmark colored according to the relative rank of its visitations at different time steps. **Right:** Landmark edges colored by goal-conditioned policy success rate at different time steps. Only edges with $\geq 80\%$ success rate are shown.

defined as reaching the next landmark within 15 steps. On top, we see that the policy is only accurate for edges near the start position during an early stage of training. Additionally, there is an incorrect, extra long edge that is most likely attributed to localization errors that will be corrected later on as the agent further trains its SF representation. On the bottom, we observe that the goal-conditioned policy improves in more remote areas after the agent has explored more of the maze and completed more training.

### E.3   Planning with Landmark Graph

Here, we look at the long-horizon landmark paths that the SFL agent plans over the graph. Examples of planned paths are shown in Figure 15. We observe that the plans accurately conform to the maze's wall structure. Additionally, consecutive landmarks in the plan are not too far apart, which helps the success rate of the goal-conditioned policy because SFS is more accurate when the start-goal states are within a local neighborhood of each other. We acknowledge that the planned paths can be longer and more ragged than the optimal shortest path to the goal. The partial observability of the environment is one primary reason for this, where there are multiple first-person viewpoints per (x, y) location. For example, two landmarks may be relatively closer in terms of number of transitions even if they appear further away on the top-down map because they both share a 30° view orientation.

## F   Mapping State Space Analysis

In this section, we offer additional analysis on Mapping State Space (MSS) and elaborate on potential reasons why the method struggles to achieve success in our environments. Our study is conducted in the context of *fixed spawn* experiments in **ViZDoom**.

### F.1   UVFA

The UVFA is central to the MSS method, acting as both an estimated distance function and a local goal-conditioned navigation policy. First, we qualitatively evaluate how well the UVFA estimates the

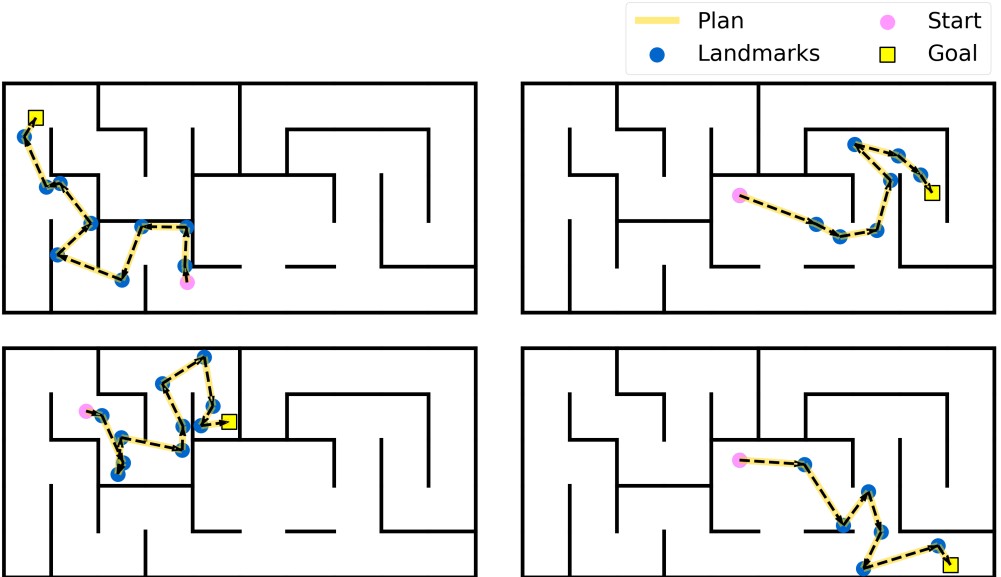

Figure 15: Examples of planned paths for various start (pink dot) and goal (yellow square) locations. The paths are formed by conducting planning on the landmark graph. These examples are taken from one of the *fixed spawn* experiments on the *Test-1* map.

distance between states by creating heatmaps similar to those in Figure 6. We use a trained UVFA to estimate the pairwise distances between a reference state and a random set of sampled states, and plot those distances in Figure 16. The top row shows the distance values for all states; the bottom row shows the distance values for states which clear the edge clip threshold $= -5$ and consequently would have edges to the reference state. We choose a edge clip threshold smaller than the one used in our experiments to reduce the number of false positives and for ease of visualization. We observe that the estimated distances are noisy overall, where many states far away from the reference state are given small distances which pass the edge clip threshold. These errors cause the landmark graph to have incorrect edges that connect distant landmarks As a result, the UVFA-based navigation policy cannot accurately travel between these distant pairs of landmarks.

We hypothesize that the UVFA is inaccurate because the learned feature space does not perfectly capture the agent's (x, y) coordinates for localization. This causes errors in the HER reward function where it may give a reward in cases when the agent has not yet reached the relabeled goal state. This is exacerbated by the perceptual aliasing problem. With this noisy reward function, the UVFA learning process becomes very challenging.

For further analysis, we re-train the UVFA with a HER reward function that is given the agent's ground-truth (x, y) coordinates. Now, the agent is only given a reward when it exactly reaches the relabeled goal state. We recreate the same UVFA-estimated distance heatmaps using this training setup, shown in Figure 17.

We see in the left column that states which pass the edge clip threshold are now more concentrated near the reference state. However, the estimated distances are still very noisy, especially in the right column where the reference state is in a distant location. We believe that learning an accurate distance function remains difficult because the features inputted into the UVFA only give a rough estimate of the agent's (x, y) position rather than capture it fully.

## F.2 Landmarks

MSS uses a set of landmarks to represent the visited state space. From a random subset of states in the replay buffer, the landmarks are selected using the farthest point sampling (FPS) algorithm, which is intended to select landmarks existing at the boundary of the explored space. Figure 18 shows the landmarks selected near the end of training in a MSS *fixed spawn* experiment. We note that when we

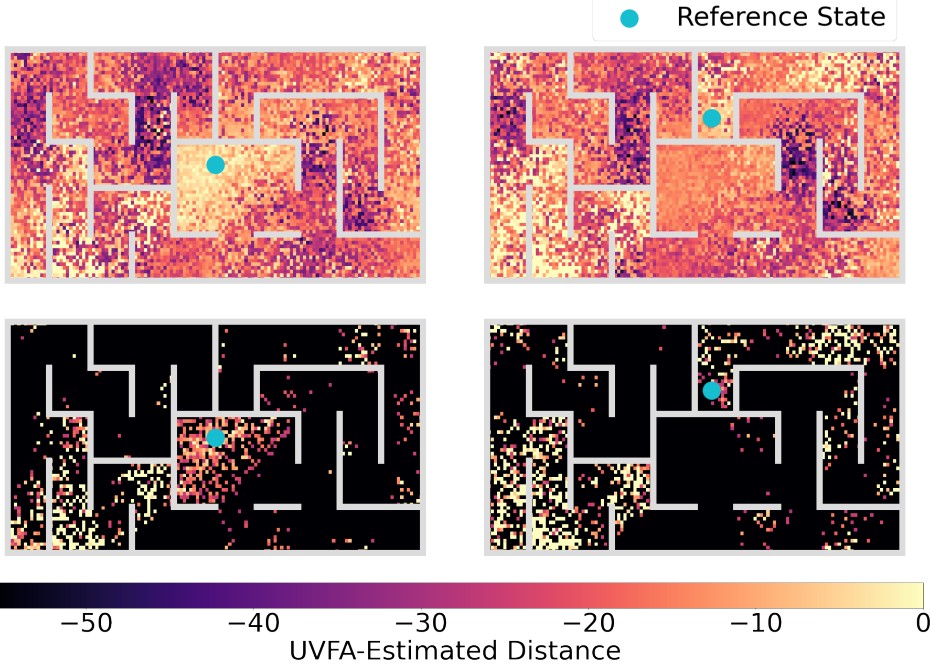

Figure 16: Distances estimated with MSS' UVFA relative to a reference state (blue dot) in the *fixed spawn* **ViZDoom** maze. The left column uses the agent's start state as the reference state while the right column uses a distant goal state as the reference state. The top row depicts all states while the bottom row shows states with distance $>= -5$, the edge clip threshold. States that do not pass the threshold, i.e. distance $< -5$ are darkened.

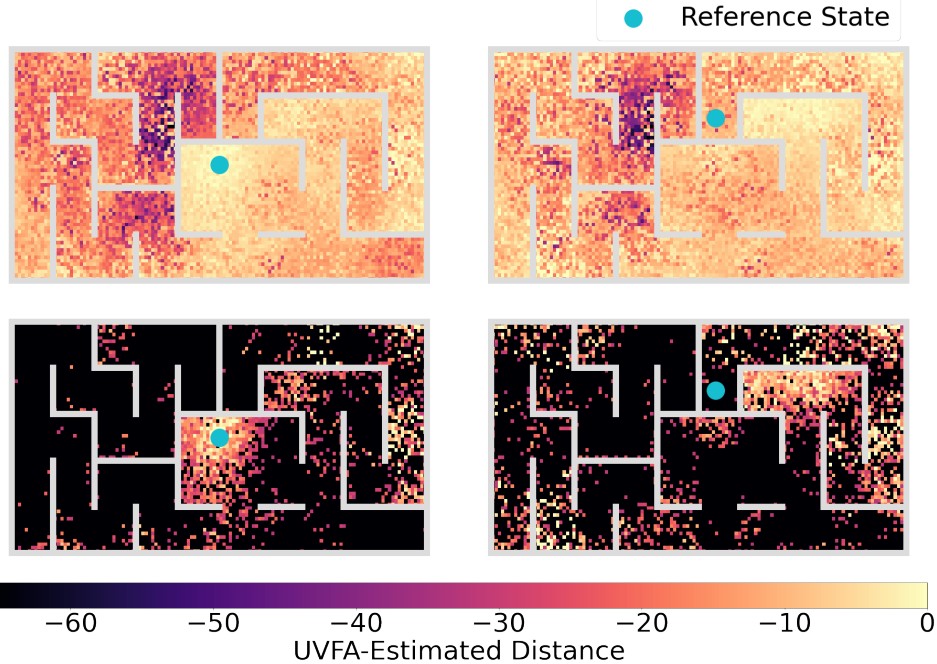

Figure 17: Distances estimated with MSS' UVFA relative to a reference state (blue dot) in the *fixed spawn* **ViZDoom** maze. We assume a similar setup as Figure 16, but the UVFA is trained using HER with ground-truth (x, y) coordinate data.

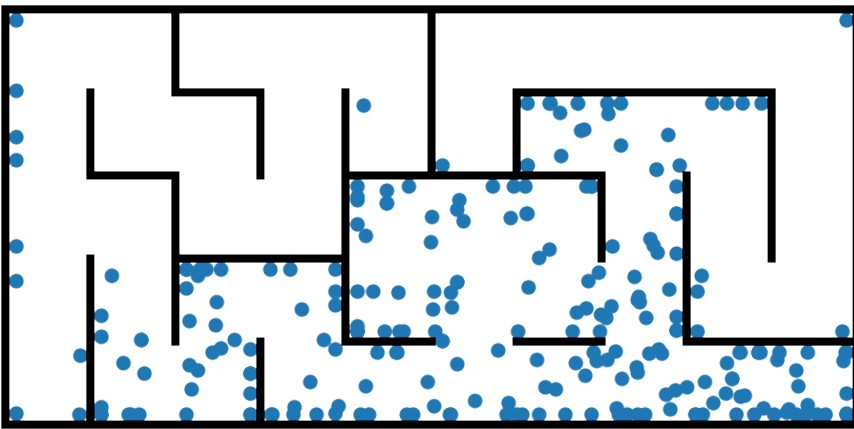

Figure 18: Landmarks selected using FPS in a MSS *fixed spawn* experiment.

increase the max number of landmarks in our experiments, the observed performance either decreases or remains the same.

The set of landmarks only partially covers the maze, thereby limiting the agent's ability to reach and further explore distant areas. Because FPS operates in a learned feature space, the estimated distances between potential landmarks can be noisy, leading to inaccurate decisions on which landmarks should be chosen. Furthermore, MSS does not explicitly encourage the agent to navigate to less visited landmarks. States in unexplored areas become underrepresented in the replay buffer and therefore, are unlikely to be included in the initial random subset of states from which landmarks are chosen.

## G   Episodic Curiosity Analysis

In this section, we conduct additional experiments regarding the Episodic Curiosity (EC) augmented SPTM and SGM baselines to better understand the benefits and shortcomings of these combined methods. This study is completed within the context of *fixed spawn* experiments in **ViZDoom**.

We run ablations of the EC + SPTM and EC + SGM baselines on the *Test-1* map within the *fixed spawn* evaluation setting. Two components of SPTM/SGM are changed in the ablations: the high-level graph, and the reachability and locomotion networks. Specifically, we vary how we collect the trajectories used to populate the graph and to train the two networks. The trajectories are generated by the following agent variations:

1. **Fixed start (FS)**: randomly acting agent that begins each episode in the *same* starting location.
2. **Random start (RS)**: randomly acting agent that begins each episode in a starting location that is sampled uniform randomly across the map.
3. **EC**: agent running the EC exploration module and begins each episode in the *same* starting location.

For populating the graph, the agents build and sample from a replay buffer containing trajectories of 100 episodes of 200 steps each, following the setup of SGM [16]. The EC agent uses a frozen checkpoint of its reachability and policy networks when collecting these trajectories. The setup for training the reachability and locomotion networks of SPTM/SGM remains the same except the agent used to generate the training data is varied between FS, RS, and EC. For the EC variant, the training data is composed of trajectories that were recorded during the training of the exploration module.

We then evaluate the underlying baselines initially described in Section 5.2, visual controller[5], SPTM, and SGM, with ablations of the methods used to populate the graph and train the networks. We report their success rates averaged over 5 random seeds in Table 10. The EC (populate graph) + FS (train networks) + SPTM/SGM baselines (underlined) are the ones reported in the main paper in

---

[5]The visual controller baseline does not use a high-level graph for planning.

| Method Used | | | Test-1 | | | |
|---|---|---|---|---|---|---|
| Populate Graph | Train Networks | Evaluation | Easy | Medium | Hard | Hardest |
| - | FS | Controller | 43% | 12% | 0% | 0% |
| - | RS | Controller | 29% | 2% | 1% | 0% |
| - | EC | Controller | 39% | 6% | 0% | 1% |
| FS | FS | SPTM | 32% | 4% | 0% | 0% |
| RS | RS | SPTM | 32% | 6% | 0% | 0% |
| EC | FS | SPTM | **48%** | **16%** | 2% | 0% |
| EC | EC | SPTM | 46% | 12% | **4%** | **4%** |
| FS | FS | SGM | 41% | 8% | 0% | 0% |
| RS | RS | SGM | 24% | 1% | 0% | 0% |
| EC | FS | SGM | 43% | 3% | 0% | 0% |
| EC | EC | SGM | 29% | 0% | 0% | 0% |
| | MSS | | 23% | 9% | 1% | 1% |
| | SFL [ours] | | 85% | 59% | 62% | 50% |

Table 10: (*Fixed spawn*) The success rates of ablations of SPTM and SGM baselines on the *Test-1* **ViZDoom** map. For each difficulty level, we bold the success rate of the best-performing ablation baseline for emphasis. We also include the success rates of MSS and our method SFL for reference.

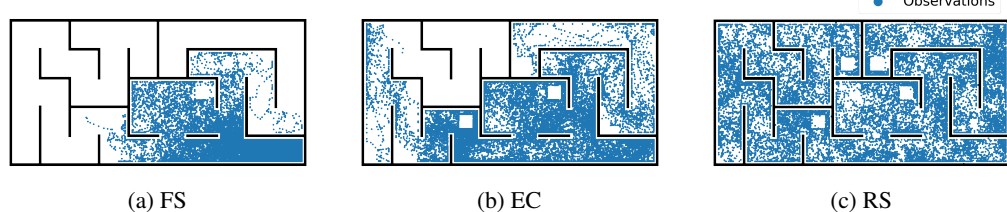

(a) FS          (b) EC          (c) RS

Figure 19: State coverage of the replay buffers built by the FS, EC, and RS agents.

Table 2. From these experiments, we make the following observations. First, using EC-generated trajectories for populating the graph can improve SPTM's performance on longer-horizon goals. This is expected as the exploration bonus from EC supports greater coverage of more distant areas of the state-space, which thereby enables graph planning to distant goals. Second, we find that the FS-generated trajectories, while limited in their coverage of distant areas, can outperform RS and EC on the Easy and Medium difficulties. We hypothesize that this is due to how *fixed spawn* start-goal pairs in evaluation share the same starting location as in training. With the FS trajectories, the training of the networks is skewed towards goals closer to the starting location. Conversely, RS trajectories suffer by having episodes start from a different location than the start location of the evaluation setting.

We also visualize the state coverage of the replay buffers used for populating the graph for the FS, EC, and RS agents in Figure 19.[6] The EC agent is able to store observations from more distant locations in comparison to the FS agent. As expected, the RS agent provides comprehensive coverage of the entire map.

---

[6]The four distinct white squares are caused by the presence of peripheral in-game objects.

## H  Additional Details: Traverse Algorithm

Here is the Traverse algorithm used for localizing the agent to the nearest landmark ($l_{\text{curr}}$) based on SFS and traversing to the next target landmark ($l_{\text{target}}$) on the landmark path with the goal-conditioned policy($\pi_l$). The procedure is repeated until either the target landmark is reached or the "done" signal is given indicating that the episode is over. This algorithm is referenced in Algorithm 1.

---

**Algorithm 3** Traverse

---

**input**  $\pi_l, l_{\text{target}}$
**output**  $\tau_{\text{traverse}}, l_{\text{curr}}$
1:  $\tau_{\text{front}} \leftarrow \emptyset$
2:  **while** $l_{\text{curr}}! = l_{\text{target}}$ or env not done **do**
3:     $(s, a, r) \sim \pi_l(\cdot; l_{\text{target}})$
4:     $l_{\text{curr}} \leftarrow \text{argmax}_{l \in L} \text{SFS}_\theta^{\bar{\pi}}(s, l)$            {localize agent}
5:     $\tau_{\text{traverse}} = \tau_{\text{traverse}} \cup (s, a, r)$
6:  **end while**
7:  **return** $\tau_{\text{traverse}}, l_{\text{curr}}$

---

## I  Societal Impact

Our Successor Feature Landmarks (SFL) framework is designed to simultaneously support exploration, goal-reaching, and long-horizon planning. We expect for our method to be applicable to real-world scenarios where an autonomous agent is operating in a large environment and must complete a variety of complex tasks. Common examples include warehouse robots and delivery drones. SFL can improve the efficiency and reliability of these autonomous agents, which offer potential benefit to human society. However, these autonomous systems may also be built for more malicious purposes such as for constructing weaponry or conducting unlawful surveillance. In general, these potential harms should be carefully considered as we begin to develop autonomous agents and pass legislation that will govern these systems.