# OpenReview forum: "Successor Feature Landmarks for Long-Horizon Goal-Conditioned Reinforcement Learning"
_NeurIPS.cc/2021/Conference — NeurIPS 2021 Poster_

### Official Review · Reviewer_RwVV · 2021-07-15

**Rating:** 7
**Confidence:** 4

**Summary:**

The paper proposes a new metric, called the successor feature similarity, that measures the similarity between two states as the dot-product between the respective successor features. The authors then use this metric to guide goal-directed exploration when learning a goal-conditioned policy by choosing a goal with the lowest count. Novel goals are stored by keeping a buffer of visited goals, where new goals are only added if they are beyond a certain distance of any prior goal. Rewards are given by the SFS between a state and a goal (where a state-only SF is computed as the action-averaged SF). The authors compare this overall method to existing graph-based GCRL methods and find that the resulting method outperforms prior work, particularly in challenging first-person maze-navigation VizDoom tasks that require moving around multiple walls.

**Limitations And Societal Impact:**

No. The authors did not discuss the limitation of this work, even though the checklist says “Yes.” The closest discussion is an acknowledgement that the method "depends on a good feature embedding to be learned." Please discuss the limitations of the work.

**Main Review:**

Overall, this paper is well written and the experiments convincingly show that the overall method outperforms prior work. The authors clearly explain how their work builds off of prior work, but also clearly presents a novel algorithm.

My main criticism is that the paper proposes a number of contributions, and the experiments would do a better job of demonstrating the impact of each individual contribution, rather than the sum of the components. For example, although the authors compare to [5], it seems that the other components of the algorithm are different. It would be useful to compare to an ablation where the SFS is replaced with the supervised learning model in [5]. The author could also compare to other metrics proposed in prior work such as in [1] which should also be discussed or [2,3,4].

Another example would be to disentangle how important the specific exploration strategy is, involving going to a frontier state and then taking random actions. Running an ablation that uses the same strategy as in SGM or SPTM would help clarify this.

Lastly, one question that is largely unaddressed is how the hyperparameters are chosen. In particular, the thresholds for adding landmarks or recording a landmark transition seem particularly important but difficult to choose.


[1] Ghosh, Dibya, Abhishek Gupta, and Sergey Levine. "Learning Actionable Representations with Goal Conditioned Policies." International Conference on Learning Representations. 2018.
[2] Ben Eysenbach, Russ R Salakhutdinov, and Sergey Levine. Search on the replay buffer:
350 Bridging planning and reinforcement learning. In Advances in Neural Information Processing Systems, volume 32, 2019.
[3] Vitchyr Pong, Shixiang Gu, Murtaza Dalal, and Sergey Levine. Temporal difference models: Model-free deep RL for model-based control. ICLR, abs/1802.09081, 2018.
[4] Srinivas Venkattaramanujam, Eric Crawford, Thang Doan, and Doina Precup. Self-supervised learning of distance functions for goal-conditioned reinforcement learning. arXiv preprint arXiv:1907.02998, 2019
[5] SPTM

**Time Spent Reviewing:**

1

---

> ### Author Response · Authors · 2021-08-10
> **Response to Reviewer RwVV**
>
> We appreciate you spending the time to review our work and give useful feedback! It is encouraging that you found the algorithm to be novel, the writing and motivation to be clear, and experiments to be persuasive. We would like to address your comments and questions below.
>
> ### 1. Comparison to ablations where SFS is replaced with another metric from related work
>
> According to your suggestion, we ran an ablation experiment where we replaced SFS with the reachability network used in SPTM [1].
>
> | Metric                          | Easy | Medium | Hard |
> | ------------------------------- | ---- | ------ | ---- |
> | SFS (Original, from paper)      |  91.6 $\pm$ 1.73% |   81.6 $\pm$ 1.31%  |  67.2 $\pm$ 2.63% |
> | Reachability Network (Ablation) |  82.8 $\pm$ 1.21% |   56.8 $\pm$ 2.57%  |  24.0 $\pm$ 1.88% |
>
> In the table above, we report the mean success rates and corresponding standard errors on the *Train* map on ViZDoom for the original SFS metric and the ablation, with the results aggregated over 5 random seeds. We observe that SFS outperforms the reachability network on all difficulty levels, indicating that SFS can more accurately predict the transition distance between states than the reachability network.
>
> ### 2. Methodological comparison to other metric learning approaches in RL literature
>
> The actionable representations for control (ARC) [2] proposed to measure the similarity of states in terms of the functional similarity of a maximum entropy goal-conditioned policy when each state is given as a subgoal input. However, they require the pre-trained maximum entropy goal-conditioned policy, which is hard to obtain in large-scale environments. The metrics proposed in [3, 4] are based on Q-functions trained with a step penalty to capture the shortest traversal time between two states. One of our baselines, Mapping State Space [5], also uses the same distance metric and we demonstrate why it struggles in large, high-dimensional environments such as ViZDoom in Appendix E. Finally, similar to [3, 4, 5], the metric in [6] is trained with a temporal distance signal via self-supervised learning. However, they have only done qualitative evaluation on high-dimensional state spaces.
>
> We will include the discussion on the metric learning in RL literatures in the revised version.
>
> ### 3. Comparison to ablations of the frontier landmarks exploration strategy
>
> We ran an additional ablation of the exploration strategy in the fixed spawn problem setting which is particularly used to evaluate exploration. The ablation samples frontier landmarks from the landmark set in a uniformly random manner rather than based on the inverse visitation counts of each landmark. This is analogous to how SGM chooses goals for their cleanup step. We note that SPTM relied on human demonstrations to build the graph and did not employ an exploration mechanism.
>
> | Frontier Sampling Strategy          | Easy | Medium | Hard | Hardest |
> | ----------------------------------- | ---- | ------ | ---- | ------- |
> | Inverse Visitation Count (Original) | 100% |   85%  | 85%  |   80%   |
> | Uniform Random (Ablation)           | 100% |   95%  | 80%  |   45%   |
>
> In the table above, we report success rates on the *Test-1* map on ViZDoom for the original exploration strategy and the ablation. The two methods are roughly comparable up until the *Hardest* difficulty, for which the original strategy outperforms the more naive ablation. Exploration is most important on *Hardest* because goals from this difficulty are furthest away from the starting state. Due to time constraints, we were only able to run 1 random seed and fewer evaluation episodes. We will provide a later update with the full results averaged over 5 random seeds.
>
> ### 4. Explanation of how hyperparameters are chosen
>
> Please refer to Common Response #2.
>
> ### 5. Limitations of this work
>
> We discuss three limitations of this work.
>
> As we briefly discussed in the conclusion section, the first limitation is the reliance on a good state embedding to learn successor features (SF) upon. If the underlying state embedding is not representative enough, the SF training will require more samples and in turn, the overall learning would be slower since our framework is built upon the representation power of SF. In future work, we foresee learning the state embedding by using a state reconstruction objective [8] or temporal distance supervision [1, 3, 4, 5, 6].
>
> The second limitation is that we used a random policy for exploring the unseen state space and training SF, which may be inefficient. We offer detailed discussion on this topic and potential directions for future work in Common Response #1.
>
> The last limitation is due to how our framework is formulated under a common goal-conditioned RL setting where we assume goals are a subset of the state space. This makes our method incompatible with tasks that cannot be defined by a state. We discuss this limitation and future research directions in our response to Reviewer V86Z’s comment #4.
>
> [1] Nikolay Savinov, Alexey Dosovitskiy, and Vladlen Koltun. “Semi-parametric Topological Memory for Navigation.” International Conference on Learning Representations. 2018.
>
> [2] Ghosh, Dibya, Abhishek Gupta, and Sergey Levine. "Learning Actionable Representations with Goal Conditioned Policies." International Conference on Learning Representations. 2018.
>
> [3] Ben Eysenbach, Russ R Salakhutdinov, and Sergey Levine. Search on the replay buffer: 350 Bridging planning and reinforcement learning. In Advances in Neural Information Processing Systems, volume 32, 2019.
>
> [4] Vitchyr Pong, Shixiang Gu, Murtaza Dalal, and Sergey Levine. Temporal difference models: Model-free deep RL for model-based control. ICLR, abs/1802.09081, 2018.
>
> [5] Zhiao Huang, Fangchen Liu, and Hao Su. “Mapping State Space using Landmarks for Universal Goal Reaching.” Advances in Neural Information Processing Systems. 2019.
>
> [6] Srinivas Venkattaramanujam, Eric Crawford, Thang Doan, and Doina Precup. “Self-supervised Learning of Distance Functions for Goal-Conditioned Reinforcement Learning.” 2020.
>
> [7] Scott Emmons, Ajay Jain, Michael Laskin, Thanard Kurutach, Pieter Abbeel, and Deepak Pathak. “Sparse Graphical Memory for Robust Planning.” Advances in Neural Information Processing Systems. 2020.
>
> [8] Tejas D. Kulkarni, Ardavan Saeedi, Simanta Gautam, and Samuel J. Gershman. “Deep Successor Reinforcement Learning.” 2016.

---

> > ### Comment · Reviewer_RwVV · 2021-08-12
> > **Re: Response to Reviewer RwVV**
> >
> > Thank you for this helpful response. It would be great to add these new results to the paper.

---

> > > ### Author Response · Authors · 2021-08-13
> > > **Re: Response to Reviewer RwVV**
> > >
> > > Yes, we will include the results from these ablation experiments in the paper revision.

---

> > ### Author Response · Authors · 2021-08-24
> > **Update: Reachability Network Ablation Experiment**
> >
> > We have finished running the 5 random seeds for the reachability network ablation experiment and have updated the table in #1 of our original response. In the table, we report the mean success rates on each difficulty level and their corresponding standard error. We observe that SFS continues to outperform the reachability network on all difficulty levels, indicating that SFS can more accurately predict the transition distance between states than the reachability network.

---

> > ### Author Response · Authors · 2021-08-31
> > **Update: Frontier Sampling Strategy Ablation Experiment**
> >
> > To more directly measure the degree of exploration achieved by each frontier sampling strategy, we decided to track state coverage. We define state coverage as the thresholded state visitation count computed over a discretized grid of agent states and report this metric as a percentage over all potentially reachable states.
> >
> > | Frontier Sampling Strategy            | State Coverage     |
> > | --------------------------------------------- | ------------------------ |
> > | Inverse Visitation Count (Original) | 79.4 $\pm$ 0.65% |
> > | Uniform Random (Ablation)           | 72.3 $\pm$ 1.90% |
> >
> > In the table above, we report the mean state coverage percentage and associated standard error computed over 5 random seeds achieved by each strategy on the *Test-1* ViZDoom map. We observe that the inverse visitation count strategy visits a higher percentage of states, indicating that it achieves greater exploration of the state space.

---

### Official Review · Reviewer_66Ma · 2021-07-16

**Rating:** 6
**Confidence:** 5

**Summary:**

This paper deals with the goal conditioned reinforcement learning problem (GCRL). It proposes an incremental approach to learning good landmark states through successor features so as to enhance exploration and in turn improve the success rate of reaching arbitrary goal states. The authors discuss learning in navigational tasks, both tabular and pixel based, where the goal state changes randomly and exploration is hard. The idea is to learn a landmark graph, each landmark being a state itself which is sufficiently distanced from other landmark states. The notion of distance is provided by the successor representation (features in the pixel case). Planning policies to reach between different landmark states alleviates the exploration issue, and thus helps in improving GCRL performance.

**Ethical Concerns:**

I do not think there any explicit ethical concerns raised by this work.

**Limitations And Societal Impact:**

Suggestions: My main suggestion would revolve around fixing the issues I point to in the 'originality' section above. I think that and more experiments in a non-navigational domain can improve this paper massively.

I do not think there any explicit negative societal impacts of this work.

**Main Review:**

Originality: The motivation of the work is solid in my opinon. However, the proposed solution lacks originality. This is my main complaint with the paper. I roughly boil them down to two main points:

- Unnecessary emphasis on 'novel' nomenclature: The paper introduces two definitions, of Successor Feature Similarity (SFS) and Successor Feature Landmark (SFL) , both claiming novelty. In my opinon, this does more harm than good, in that it hampers the readibility/understanding of the paper. Both these ideas are not new in any sense, and I claim that any reader with prior knowledge of this area would recongnize this instantly. Successor Feature Similarity (SFS) aims to be a distance metric based on the successor representation. This idea dates back to the Proto-value functions work, or at the very least, is as old as the learning eigen options from the successor representation paper by Machado et. al. Similarly, the idea of landmark states for better exploration probably dates back to the multi-value functions paper by Moore et. al.

- (Lack of) discussion to previous work: The options literature has looked at the problem of navigation extensively for decades, and yet the authors do not provide a discussion on how these ideas are similar/different from those in the options literature. Successor options by Ramesh et. al. seems to be the closest to this paper, where a clustering scheme (over the successor representation) is used to identify landmark states. I believe the graph based solution, based on looking at distances between states using the SFS metric is doing exactly the same. The authors do not discuss how exactly their approach is beneficial over the clustering scheme. For example, one benefit could be that modelling the states as a graph explicitly allow for better identification of fringe nodes, thus aiding in better directions to explore. If this is the case, an simple experiment to show this would really help in highlighting the main contributions of the paper.

Quality: The overall quality of the work is quite high. The experiments are well executed and observations are nicely reported.

Clarity: The paper reads quite clearly overall. The flow of content is pretty nice as well.

Significance: Overall, I think the main contribution of this work is not the main ideas but the results on pixel based navigation, which I believe are really impressive and shows that these and similar ideas are easily scalable. I do not gain any conceptual insight from this, nor do I think the paper is aiming to provide any.

Questions: My primary question is regarding the similarity to the idea of doing clustering over the built SF to get the landmark states and then learning policies to reach these (also called options). It would be really helpful if the authors can discuss the exact differences and similarities.

**Time Spent Reviewing:**

5 hrs

---

> ### Author Response · Authors · 2021-08-10
> **Response to Reviewer 66Ma**
>
> Thank you for spending the time to review our work and provide valuable feedback! We are glad that you found the motivation, quality of work, and presentation to be strong. It is also encouraging that you thought the results on the long-horizon goal-conditioned RL (GCRL) problem in high-dimensional environments were impressive. We would like to address your concerns below:
>
> ### 1. The Successor Feature Similarity (SFS) metric and Successor Feature Landmarks (SFL) framework are not novel ideas.
>
> We agree that individual components of our framework share similar ideas with those proposed in prior literature. However, we claim that our main conceptual contribution is how we can build the *entire framework*, which includes the goal-conditioned policy, landmark graph, and planner, *solely upon the SFS metric*. We note that this makes the overall learning faster (via knowledge sharing) and more stable (single learning component).
>
> We also claim that our form of the SFS metric is novel. While Machado et al. 2018 [1] appears to propose a very similar form to SFS in its eigenpurpose reward functions, these eigenpurposes are related to latent directions of the state space while SFS serves as a distance metric.
>
>
> * More specifically, Machado et al. 2018 [1] define eigenpurposes as $r_{i}^e (s, s’) = e^{\top}(\phi(s’) - \phi(s))$ (Eq. 1 in [1]). The first term, $e$, is an eigenvector of the successor representation (SR), which is equivalent to a proto-value function [2] (PVF) up to scale (Section 3.2, [1]). A PVF captures a latent direction out of a basis set of directions that represent the state space. The latter term, $\phi(s’) - \phi(s)$, is the feature displacement resulting from the agent’s transition. Thus, the dot product between these two is the projection of the agent’s displacement onto the latent direction, and the eigenpurpose reward function measures the agent’s **progress towards that latent direction**.
>
>
> * On the other hand, our SFS form is the dot product between the (normalized) successor features (SF) of the sub-goal and of the current state. This form is the cosine similarity, or the **distance between the two SF vectors** on the unit sphere, and intuitively measures distance between two states based on their expected future state traversity. We also show that our form enables us to instantly compute a goal-conditioned Q-value function from SFS (and hence policy) *without any additional policy learning* (see Section 4.3). Thus, to the best of our knowledge, our SFS form is novel and its specific form provides a strong benefit.
>
> We agree with you that the idea of using landmark states for better exploration is not new and we only claim minor novelty in this aspect. That said, we would like to clarify how our work differs from prior work on using landmarks for exploration. Since multi-value-functions [3], numerous papers [4, 5] have identified landmarks as states which enable access to distinct regions of the state space and thus, improve exploration. In comparison with the most closely related work, Mapping State Space (MSS) [6], we note two main differences. MSS populates landmarks using farthest point sampling while we progressively grow the landmark set based on novelty and state coverage (see response #1 to ReviewerV86Z for details). Also in exploration, MSS chooses a random landmark as an exploration target while ours chooses based on visitation count (i.e., novelty).
>
> We will include the above discussion in the revised paper.
>
> ### 2. There is a lack of discussion of options literature.
>
> We agree that the options literature includes ideas that are analogous to our framework. Our method is an instance of graph-based planning, and thus adopts the hierarchical policy form which has a close connection to the options framework. Our framework also lies more in the category of goal-conditioned RL, where a *shared* goal representation is used.
> At the meta-policy level, our framework uses the graph planner to *compute* the next landmark (or subgoal, option) to reach while the options framework *learns* a policy that chooses over the set of options.
>
> At the low-level policy level, the options framework defines each option in terms of the initiation set, termination set and option policy. In the graph-based planning framework, each edge in the graph can be seen as an option, where the starting node, ending node, and the goal-conditioned policy transitioning from starting node to ending node can be seen as the initiation set, termination set, and option policy, respectively. In the options framework, each option is often learned separately, whereas ours computes *a single goal-conditioned policy* for all the edges.
>
> To the best of our knowledge, the most closely related works among the options literature are Eigenoptions [1] and Successor Options [7]. We discuss these works in Comment #1 above and Comment #3 below. We will also include the comparison with the options literature in our revised version.
>
> ### 3. Comparison to Successor Options [8]
>
> Successor Options presents an options framework for discovering sub-goals and learning options via the successor representation. There exists similarities and differences due to the fact that SFL resides under the graph-based planning and GCRL frameworks which we have discussed above in Comment #2.
>
> Beyond that level of comparison, we note a key difference in how the lower-level policy is obtained. In SFL, we do not perform any policy learning and can instantly compute a goal-conditioned policy from SFS (see Section 4.3). This policy can be used for any sub-goal. On the other hand, in Successor Options, the agent has to learn a separate option policy from the pseudo-reward function for each individual sub-goal. Similar to Machado et al. 2018 [1], Successor Options only qualitatively evaluated their method in the function approximation setting.
>
> ### 4. What are the benefits of using SFS to identify landmark states over the clustering scheme used in Successor Options [7]?
>
> The primary benefit of our approach over the clustering scheme stems from how our landmark set progressively grows. Since we progressively build the landmark set, we maintain all previously added landmarks. Then, we are able to utilize many useful notions such as 1) when landmarks were added, 2) how many times the agent has visited (or been localized to) each landmark, and 3) what transitions have occurred between landmarks, etc to improve the connectivity quality of the graph. In comparison, the clustering in [7] is unable to achieve this effect because the landmark set is rebuilt every few iterations (see Algorithm 1 in [7]).
>
> A secondary benefit lies in which landmarks are chosen. Our approach aims to minimize the distance between chosen landmarks as controlled by the add threshold while clustering selects landmarks which are most representative of states in topologically distinguishable regions. We agree with your example benefit - in our approach, we add landmarks that are far away from existing landmarks, making them likely to lie on the edge of the explored state space by nature. This can then improve exploration via our frontier strategy.
>
> We ran an additional experiment on MiniGrid (FourRoom) to empirically compare our progressive construction approach versus clustering. We used the same landmark limit and training data for both methods. The average pairwise geodesic distance between landmarks was 6.72 +/- 0.43 for our method and 5.72 +/- 0.40 for clustering, showing that our method’s landmarks were slightly more spread out than those identified by clustering.
>
>  [1] Marlos C. Machado, Clemens Rosenbaum, Xiaoxiao Guo, Miao Liu, Gerald Tesauro, and Murray Campbell. “Eigenoption Discovery through the Deep Successor Representation.” International Conference on Learning Representations. 2018.
>
>  [2] Sridhar Mahadevan. “Proto-Value Functions: Developmental Reinforcement Learning.” International Conference on Machine Learning. 2005.
>
>  [3] Andrew W. Moore, Leemon C. Baird, and Leslie Kaelbling. “Multi-Value-Functions: Efficient Automatic Action Hierarchies for Multiple Goal MDPs.” International Joint Conference on Artificial Intelligence. 1999.
>
>  [4] Özgür Şimşek and Andrew G. Barto. “Using relative Novelty to Identify Useful Temporal Abstractions in Reinforcement Learning.” ACM. 2004.
>
>  [5] Özgür Şimşek, Alicia P. Wolfe, and Andrew G. Barto. “Identifying Useful Subgoals in Reinforcement Learning by Local Graph Partitioning.” International Conference on Machine Learning. 2005.
>
>  [6] Zhiao Huang, Fangchen Liu, and Hao Su. “Mapping State Space using Landmarks for Universal Goal Reaching.” Advances in Neural Information Processing Systems. 2019.
>
>  [7] Rahul Ramesh, Manan Tomar, and Balaraman Ravindran. “Successor Options: An Option Discovery Framework for Reinforcement Learning.” International Joint Conference on Artificial Intelligence. 2019.

---

> > ### Comment · Reviewer_66Ma · 2021-08-13
> > **Re: Response to Reviewer 66Ma**
> >
> > Thank you for the detailed reply and for conducting additional ablations in repsonse to this review and to the one by reviewer RwVV. I largely agree with the discussion about the options literature and the primary benefit with respect to [7]. I hope this can be included as part of the related work.
> >
> > - I am a bit stuck at the point about computed the sub-goal policy instantly as opposed to learning option policies. I do not see a mention of how the policy is instantly computed in Sec 4.3. I only see a mention of the Q values there. Is it the case that since we are in the discrete action setup, the action at any given state is computed just by the max operation? If so, how must one tackle the continous action case?
> >
> > - Regarding the second bullet to point 1 in your response, it looks like the reward definition in [7] is much closer to the SFS metric as compared to the one proposed in [1]. In line with that, are you able to run the clustering version for the entire setup (instead of only comparing how spread out the goals are)? Or are there clear issues in implementing that idea?
> >
> > Overall, it seems that the main idea is similar to a lot of prior work and thus this should be well noted before the exact mechanics of the framework are discussed. Also, more detailed comparisons, especially to the clustering technique in [7] should be discussed in the later sections, i.e. when introducing the SFS metric. I also believe there should be less emphasis on how this is an entirely new framework to solve vision-based navigational domains (which is how it felt like when reading the paper) and instead more focus should go on the empirircal contributions. Hopefully, adding the already conducted ablations can help improve on this aspect.

---

> > > ### Author Response · Authors · 2021-08-19
> > > **Re: Re: Response to Reviewer 66Ma**
> > >
> > > Thank you for your response and valuable comments! We are pleased that you agree with our comparisons to the options literature and to Successor Option’s [1] clustering scheme. We will include this discussion in the paper revision. We would like to address your comments and questions below.
> > >
> > > ### 1. It seems that the main idea is similar to a lot of prior work. I also believe there should be less emphasis on how this is an entirely new framework to solve vision-based navigational domains (which is how it felt like when reading the paper).
> > >
> > > While we disagree that our main idea itself is not novel (as discussed in #1 and #3 of our previous response), we agree that graph-based planning frameworks have existed in prior work and we do not claim this concept as our contribution. We will update our discussion of prior work to acknowledge this more clearly. To explicate what is and isn’t our novelty, we will add a paragraph that consolidates our specific novelty claims (which are reiterated below).
> > > 1. We use a single self-supervised learning component, successor features (SF), to build all the components of a graph-based planning framework. We claim that this enables knowledge sharing between each module in the framework, and stabilizes the overall learning (see #1 in our previous response).
> > > 2. We introduce the SFS metric, which serves as a distance estimate and enables computing a Q-value function *without further learning*. From this, we can simply use the argmax operation to instantly obtain a goal-conditioned policy. Successor Options [1] used a reward function similar to ours, but performed *additional policy learning* to obtain separate option-policies for maximizing their reward.
> > >
> > > ### 2. Is it the case that since we are in the discrete action setup, the action at any given state is computed just by the max operation? If so, how must one tackle the continuous action case?
> > >
> > > Yes, you are correct in that we are in the discrete action setup and can select actions at any given state by applying the argmax operator over the Q-value function, thus instantly obtaining the goal-conditioned policy. In the continuous action case, we can learn the goal-conditioned policy from the Q-value function by using a compatible algorithm such as DDPG [2], but extending SF learning to a continuous action setup is beyond the scope of this work and is left for future work.
> > >
> > > ### 3. Are you able to run the clustering version for the entire setup?
> > >
> > > We are currently implementing the clustering scheme for the entire setup and will report the results of the experiments once finished. We will include detailed comparisons to the clustering scheme as well as the other ablation studies in the paper revision.
> > >
> > > [1] Rahul Ramesh, Manan Tomar, and Balaraman Ravindran. “Successor Options: An Option Discovery Framework for Reinforcement Learning.” International Joint Conference on Artificial Intelligence. 2019.
> > >
> > > [2] Timothy P. Lillicrap, Jonathan J. Hunt, Alexander Pritzel, Nicolas Heess, Tom Erez, Yuval Tassa, David Silver, and Daan Wierstra. “Continuous control with deep reinforcement learning.” International Conference on Learning Representations. 2016.

---

> > > > ### Comment · Reviewer_66Ma · 2021-08-22
> > > > **Thanks!**
> > > >
> > > > Thank you for the detailed reply. I am quite satisfied with the clarifications above and therefore am updating my score to 6. I hope the discussion we had thus far and the ongoing experiments can be included in the revised paper. It would also be good to mention that for the continous action case, we would need to learn individual policies.

---

> > > > > ### Author Response · Authors · 2021-08-24
> > > > > **Re: Thanks!**
> > > > >
> > > > > We are glad that you found the discussion to be fruitful and appreciate you taking the time to have this discourse with us. We will incorporate the discussion and the additional conducted experiments in the paper revision. We will also include the note about learning individual policies in the continuous action setting.

---

> > > > ### Author Response · Authors · 2021-08-26
> > > > **Update: Clustering Ablation Experiment**
> > > >
> > > > We report the results of the ablation experiment where we identify landmarks using the clustering scheme described in Successor Options [1].
> > > >
> > > > | Landmark Identification Scheme | Success Rate |
> > > > | ------------------------------------------- | ------------------- |
> > > > | Progressive Construction (Original) |  75.0 $\pm$ 14.6% |
> > > > | Clustering (Ablation) |  35.6 $\pm$ 18.8% |
> > > >
> > > > In the table above, we report the mean success rates and corresponding standard errors on Three-Room MultiRoom. These results are aggregated over 5 random seeds. We observe that our method for building the landmark set more than doubles the success rate achieved by the clustering scheme. We attribute this outperformance to how our method progressively builds up the landmark set. This enables us to track useful metadata about the landmarks such as which transitions have occurred between landmarks, and we can use this information to improve the connectivity quality of the graph (more details in #4 of our original response). In contrast, the clustering scheme rebuilds the entire landmark set every few iterations and thus, is unable to maintain this type of beneficial information.
> > > >
> > > > [1] Rahul Ramesh, Manan Tomar, and Balaraman Ravindran. “Successor Options: An Option Discovery Framework for Reinforcement Learning.” International Joint Conference on Artificial Intelligence. 2019.

---

### Official Review · Reviewer_YnN2 · 2021-07-20

**Rating:** 7
**Confidence:** 3

**Summary:**

This paper presents Successor Feature Landmarks (SFL), a new graph-based planning framework based on successor features for long-horizon goal-conditioned RL (GCRL). The key idea is to use successor features to define a new metric, Successor Feature Similarity (SFS), to estimate the distance between two states or state-action pairs in high-dimensional environments, such as 3D visual environments. By leveraging SFS, a landmark-based graph is built to represent the explored state space. Both guided/directed exploration and random exploration are deployed to help the agent traverse between landmarks and explore unseen states of the environment respectively. Combining all the components, SFL is more efficient and scalable than previous graph-based methods for long-horizon GCRL.

**Limitations And Societal Impact:**

The authors discussed the limitation in Section 6 and social impact (both positive and negative) in Appendix H.

**Main Review:**

Measuring the distance between states using similarity between the corresponding successor features is a new and interesting idea. This design not only considers the actual transition dynamics in the environment but also makes it feasible to estimate distances in high-dimensional state spaces. The resulting framework SFL with landmark-based non-parametric graph and goal-conditioned exploration is well designed and presented in the paper. Figures 1, 2 and Algorithms 1, 2 are extremely helpful to illustrate the details of the framework in my opinion. Empirically, the authors have included a set of extensive experiments, and the performance of SFL looks quite promising for long-horizon GCRL.

Overall, the paper is well-written and organized, and the proposed method is technically sound. I am generally satisfied with the submission, but I listed some items below which I believe if revised could further improve the overall clarity of the paper.

(1)	Is the successor feature $\psi(s)$ or $\psi(s, a)$ normalized? I don’t think this is mentioned in the paper, but I believe it is important to clarify this. Accordingly, is the distance measure SFS bounded (eqn. (5) or (6))?

(2)	Line 201-202 says “If we let $w$ be $\psi^{\bar{\pi}}(g)$, … being equal to the SFS between $s$ and $g$”. However, if strictly following eqn.(3) and letting $w$ be $\psi^{\bar{\pi}}(g)$, eqn.(8) seems not correct to me. Conditioned on policy $\pi$, $Q^{\pi}(s, a, g) = \psi^{\pi} (s, a)^T \psi^{\bar{\pi}}(g)$, which is then not equal to the SFS between $s$ and $g$ that follows a random policy. I could understand the idea here, but probably revising the wording a bit can improve the clarity. In addition, how does the local goal-conditioned policy $\pi_l$ be derived? I assume that $\pi_l$ is extracted from $ Q^{\pi}(s, a, g)$ rather than separately learned. It could be helpful to explicitly state the derivation of $\pi_l$ and how actions are sampled from $\pi_l$ in Section 4.3.

(3)	In Algorithm 2, line 2, do you actually mean for $s \in \tau$ do? Line 11, do you also need to check if $l_{prev} \neq \emptyset$ and $l_{prev} \neq l_{curr}$? In addition, any suggestions on how to select a reasonable combination of parameters $\delta_{add}, \delta_{local}$ and $\delta_{edge}$ in practice when applying SFL to a new environment (not included in the paper)?

(4)	Is there any limit on the total number of landmarks $|L|$ tracked in the graph? I believe it could be interesting to have some mechanisms to remove or update/replace landmarks in the set $L$. SFS is being constantly updated during learning. Even if $l$ and $l’$ are very close to each other, $l’$ can be easier to reach than $l$ when used as a goal conditioning. Therefore, it could be interesting to replace $l$ with $l’$ in the graph.

(5)	Experiments on MultiRoom in Figure 5 are of high variance. It could be helpful to add more elaboration on this part to explain the variance.

(6)	Exploration in unseen state space is conducted via random action sampling, which is in general quite inefficient. I understand that this is needed in the framework to learn SF, but it could be another limitation of the current algorithm.



**Time Spent Reviewing:**

3.5

---

> ### Author Response · Authors · 2021-08-10
> **Response to Reviewer YnN2**
>
> Thank you very much for reviewing our work and providing helpful comments! It is exciting that you found the proposed SFS metric to be new and interesting, the SFL framework to be well-designed, and the empirical results of SFL to be strong for long-horizon goal-conditioned RL. We are also pleased to hear that Figures 1, 2 and Algorithms 1, 2 were useful in aiding your understanding of the framework. We would like to address your comments below:
>
> ### 1. Is $\psi(s)$ or $\psi(s, a)$ normalized? Is SFS bounded?
>
> $\psi(s, a)$ is normalized such that $ \|\|\psi(s, a)\|\|_2 = 1$. Consequently, SFS is the cosine similarity between the SF of two states and is bounded between -1 and 1. We will clarify this point in the next revision.
>
> ### 2. Equation 8 comparing the Q-value to SFS seems incorrect. In addition, how is the local goal-conditioned policy derived?
>
> Thank you for pointing this out. We will clarify that we choose for the $\pi$ in $Q_{\pi}(s, a, g)$ to also be $\bar{\pi}$. This choice makes $Q_{\pi}(s, a, g)$ equivalent to a slightly modified version of SFS between a state-action $(s, a)$ and a state $g$. The local goal-conditioned policy $\pi_l$ then uses this goal-conditioned Q-value function to select actions in an epsilon-greedy manner.
>
> ### 3-1. In Algorithm 2, do you mean “for $s \in \tau$ do” on line 2?
>
> Yes, we will make this correction in the next revision. We had meant “for $s : \tau$” to mean “to iterate **in order** over each state $s$ in the trajectory $\tau$”, but will change it back to “$s \in \tau$” since it is a more standard math notation.
>
> ### 3-2. Do you need to check for $l_{\text{prev}} \neq \emptyset$ and $l_{\text{prev}} \neq l_{\text{curr}}?$ on line 11?
>
> Yes, we also need that check on line 11 - thank you for spotting this detail. We will include it in the next revision.
>
> ### 3-3. Do you have any suggestions for selecting values for the threshold parameters when applying SFL to a new environment?
>
> Please refer to our Common Response #2.
>
> ### 4. Is there a limit on the total number of landmarks? It would also be interesting to include mechanisms to update and replace landmarks.
>
> In MiniGrid, we set the limit to 30 landmarks given the environment’s smaller size. Empirically, the add threshold $\delta_{\text{add}}$ implicitly prevents the number of landmarks from saturating to the limit on smaller maps. However, when we do hit the limit, we will replace a landmark $l$ with a new landmark $l’$ in the following manner. First, we identify two landmarks $a, b$ such that $\underset{a, b \in L}{\operatorname{argmax}}SFS(a, b)$. Then, we select the landmark to replace $l = \underset{x \in a, b}{\operatorname{argmax}}SFS(x, l’)$. Intuitively, we identify the two landmarks that are most similar to each other and therefore have the most redundancy in state coverage. Then, out of those two, we replace the one that is most similar to the proposed new landmark.
>
> In ViZDoom, we currently set a limit of 2500 landmarks for computational efficiency purposes, but the number of landmarks also never empirically reached this limit due to the add threshold. For example, we ended up with an average of 2341 landmarks on the Train map and 404 landmarks on the Test-1 map.
>
> We agree that it would be interesting to further study different landmark replacement mechanisms, especially when under tighter computational constraints.
>
> ### 5. Experiments on MultiRoom in Figure 5 have high variance. It would be useful to elaborate on this.
>
> Yes, the MultiRoom plots shown in Figure 5 appear to have high variance (i.e., larger and more frequent fluctuation) in success rate over the course of training compared to other plots. This is because we evaluated the agent more frequently on MultiRoom in comparison to other tasks, and did not perform any smoothing. In the paper revision, we will update the plots by matching the evaluation and visualization setting.
>
> ### 6. Exploration in the unseen state space is conducted via random action sampling, which can be inefficient. This may be another limitation of the current algorithm.
>
> Please refer to our Common Response #1.

---

> > ### Comment · Reviewer_YnN2 · 2021-08-30
> > **Thank you**
> >
> > Thanks very much for the detailed response. My questions have been addressed. Please also add the additional clarifications in the revision, and I believe they are helpful to the readers.

---

> > > ### Author Response · Authors · 2021-08-31
> > > **Re: Thank you**
> > >
> > > We will include the clarifications in the paper revision. Thank you again for your time and constructive feedback.

---

### Official Review · Reviewer_V86Z · 2021-08-02

**Rating:** 8
**Confidence:** 5

**Summary:**

Summary:
=============================
This paper aims to solve the long-term goal-conditioned planning problem by tackling strategies to explore state-goal pairs. The proposed methods improve the exploration performance by formulating successor features as landmark graphs and in exploration, use three key steps: (1) select the frontier graph node; (2) use a random policy to explore; (3) extend the landmark graph by a successor feature similarity metric.

Overall, this paper is well written, with methodology clearly explained and evaluation done in detail. So I recommend a clear acceptance.

**Limitations And Societal Impact:**

Yes.

**Main Review:**

Strength:
=========================
The proposed method is convincible for me, both in intuitions and experiments. Regarding intuition, successor features represent the local traversity of states, which describes the system dynamics using value predictions. So using SF to represent landmarks and to explore the state-goal pairs is reasonable. Regarding experiments, various baselines, including graph-based planning baselines  (SGM, SPTM) and UVFS based baseline (MSS) are implemented and compared, showing the proposed method’s supreme performance. Results are not surprising, but I appreciate the author’s effort for due diligence.

Minor concerns:
=========================
However, here are some minor concerns, with the hope to help improve the paper:

(1.) While I agree using SF similarity metric is a good one to measure how novelty a state is since SF only considers the environment’s system dynamics, I am still wondering why? Considering SF represents the expected state traversity, the dot product (Page 5, equation 4) between two SF vectors is projecting on SF on the other. Then what does such projection mean?

(2.) While I also agree using a random policy as a fixed policy in your SF training is a good choice since as explained, it is uniformly random so that only dynamic of environment will dictate which states the agent will visit, but this will also make the SF pre-training hard. For example, it may require a huge data corpus in order to accurately approximate the SF since the use of uniformly random policy to explore and to represent SF. Any explanations for that?

(3.) While I agree as well using the frontier of the graph node is a good choice to set as the exploration anchor point, but compared to graph sparsity [SGM] and graph Laplacian methods, this selection ignores the interconnection nature inside a graph, will that give a clue to better improve the frontier selection?

(4.) Goal-conditioned tasks, as shown in the paper demoed in navigation tasks, assume the goal can be defined as a target state. This assumption enables exploring state-goal pairs by adding new states as the new goals. However, many RL tasks can not represent the goal as a reachable state. For example, the goal is to survive in VizDoom, which has no relationship with any state reaching. Or the goal is to set some preference to some states while reaching the goal. As you can see, the SF landmark structure introduced here is not a general framework. There are other general methods [Bagaria et al. 2021], however, that can not handle high-dimensional state-action space. Expecting to see more works done here.

Reference:
Bagaria, Akhil, Jason K. Senthil, and George Konidaris. "Skill Discovery for Exploration and Planning using Deep Skill Graphs." International Conference on Machine Learning. PMLR, 2021.


**Time Spent Reviewing:**

3 hours

---

> ### Author Response · Authors · 2021-08-10
> **Response to Reviewer V86Z**
>
> Thank you for taking the time to review our work and provide helpful comments! It is encouraging that you found the writing to be clear and the method to be compelling from both an intuitive and experimental standpoint. We would like to address your comments and questions below:
>
> ### 1. “Why is SF similarity (SFS) a good novelty measure? What does such a projection mean?”
>
> First, we would like to clarify that we use SFS as a distance metric, and not as the novelty metric itself. Instead, the novelty is measured in combination with other mechanisms. Specifically, the novelty is measured in two ways: visitation count and state coverage. First, we maintain the visitation (i.e., localization based on SFS) count for each landmark where lower count means higher novelty. Second, we consider a state that has low SFS (i.e., large distance) to *all* the landmarks as a novel state since such a state is far from the explored region. We note that both measures are built upon the SFS metric, so the quality of our novelty measure depends on the capacity of SFS as a distance metric. We believe that SFS is a good distance metric because it is formulated on SF, an augmented representation of state which takes into account not only the state, but also the transition dynamics to capture the topological structure of the environment.
>
> To interpret SFS’s meaning as a dot product, we first follow your consideration of SF as a vector representing the expected state traversity. Since SF vectors are made to have unit norm, SFS is defined as the cosine of the angle between the two vectors. We can then interpret SFS to measure the degree to which two expected state traversity distributions align.
>
> ### 2. Using a uniform random policy for SF training can be inefficient.
>
> Please refer to our answer in Common Response #1.
>
> ### 3. SFL only considers the novelty of state but ignores the “interconnection between states” (e.g., transitions) for choosing the exploration anchor point.
>
> We understand your comment as follows: our method uses newly added (frontier) landmarks as exploration anchor points, and we only consider *state novelty* with respect to the SFS metric when adding landmarks to the graph. However, other methods (SGM, graph Laplacian) also consider the *state coverage*, which measures the degree to which landmarks represent  an aggregation of neighboring states in terms of the goal-conditioned value function or *transition dynamics*.
>
> We claim that using SFS to add landmarks also takes into account the landmarks’ degree of state coverage. First, we consider SF to represent the expected state traversity when starting from a state and observe that we add a state as a landmark if the SFS between the state and the existing landmarks is small. If SFS values are small, then the expected state traversities of the state and the existing landmarks are largely different. Therefore, we will add a state as a landmark if starting from this state, the agent is expected to visit states that were not covered by (or visited from) existing landmarks. That being said, more explicitly considering connectivity of the state space in our exploration strategy could be an interesting future research direction.
>
> ### 4. SFL cannot handle a goal that is not representable as a single reachable state. Wondering if SFL can be extended in a similar way to [Bagaria et al. 2021].
>
> We agree that our framework is not compatible in settings where the goal cannot be represented by any reachable state. Our work is formulated within a common goal-conditioned RL setting where the goals are a subset of the state space [1, 2, 3] or the entire state space [4, 5].
>
>  [Bagaria et al. 2021] [6] extends the graph such that nodes can be a set of states, which enables representing goals that are not defined by a single state. We may interpret a landmark in our framework as representative of the set of states in a local neighborhood around it as defined by some metric space. However, due to our design of the metric space to follow geodesic distance, we are unable to define an arbitrary set of states that are not geodesically clustered as our goal landmark.
>
> A future research idea would be to learn a metric space such that a set of goal states becomes clustered in the learned metric space even if they are not geodesically close to each other. Then, this set of goal states can once again be represented by a single landmark state which is compatible with our framework. Learning these different metric spaces would be interesting future work.
>
>
>  [1] Leslie Kaelbling. Learning to achieve goals. International Joint Conference on Artificial Intelligence. 1993.
>
>  [2] Tom Schaul, Daniel Horgan, Karol Gregor, and David Silver. “Universal value function approximators” International Conference on Machine Learning. 2015.
>
>  [3] Zhiao Huang, Fangchen Liu, and Hao Su. “Mapping State Space using Landmarks for Universal Goal Reaching.” Advances in Neural Information Processing Systems. 2019.
>
>  [4] David Warde-Farley, Tom Van de Wiele, Tejas Kulkarni, Catalin Ionescu, Steven Hansen, and Volodymyr Mnih. “Unsupervised Control Through Non-Parametric Discriminative Rewards.”  International Conference on Learning Representations. 2019.
>
>  [5] Ofir Nachum, Shixiang Gu, Honglak Lee, and Sergey Levine. “Data-Efficient Hierarchical Reinforcement Learning.” Advances in Neural Information Processing Systems. 2018.
>
>  [6] Akhil, Bagaria, Jason K. Senthil, and George Konidaris. "Skill Discovery for Exploration and Planning using Deep Skill Graphs." International Conference on Machine Learning. PMLR, 2021.

---

### Author Response · Authors · 2021-08-10
**Common response to the Reviewers**

We appreciate the reviewers for their helpful comments. Please see individual responses below, and here are responses to common questions.

### (Reviewers V86Z and YnN2) 1. Using a uniform random policy to explore the unseen state space and train SF is inefficient.

We believe that the inefficiency of using the random policy is largely mitigated by the use of the goal-conditioned policy and planner in moving the agent to the edge of the explored state space. The random policy is then deployed to explore only the local neighborhood at this frontier region for a short horizon. Under this framework, the agent is able to visit a diverse set of states in a relatively efficient manner. In our experiments on the large ViZDoom maps (see Section 5.4), we demonstrate that this strategy is sufficient for learning a SF representation that ultimately outperforms the baselines. In fact, our framework can use **any** fixed policy for local exploration; we chose the uniform random policy due to its generality. A future research direction would be to incorporate an exploration policy that can be quickly learned to improve the local exploration around frontier.

### (Reviewers YnN2 and RwVV) 2. How are the $\delta_{\text{add}}, \delta_{\text{local}}, \delta_{\text{edge}}$ hyperparameters selected? Do you have suggestions on how to select them in a new environment?

We tune $\delta_{\text{add}}$ and $\delta_{\text{local}}$ on the *Train* map on ViZDoom, performing grid search over the values of [0.70, 0.75, 0.80, 0.85, 0.90] for $\delta_{\text{add}}$ and [0.70, 0.80, 0.9, 0.95] for $\delta_{\text{local}}$ as described in Appendix B.4. We then use the best performing values on all other ViZDoom experiments. We found that our method performed well under a range of values for $\delta_{\text{add}}$ and $\delta_{\text{local}}$. For example, a seed-controlled experiment on the *Train* map on ViZDoom for random spawn showed that varying $\delta_{\text{add}}$ to have values of 0.7, 0.8, and 0.9 respectively resulted in 41%, 46%, and 47% success rates on *Hard* tasks. Likewise, varying $\delta_{\text{local}}$ to have values of 0.8, 0.9, 0.95 resulted in 24%, 46%, and 44% success rates. For comparison, SGM, the best-performing baseline, had a 26% success rate on *Hard* tasks.

$\delta_{\text{edge}}$ is automatically computed in ViZDoom as the median number of landmark transitions computed over all possible edges (see Appendix C.2). In a smaller scale environment such as MiniGrid, $\delta_{\text{edge}}$ is simply set to 1 meaning if there is a repeated landmark transition, then we should add that edge.

We had a similar future idea for automatically computing $\delta_{\text{add}}$ and  $\delta_{\text{local}}$. For $\delta_{\text{add}}$, we would first choose how far apart we want our landmarks to be in terms of number of actions and call this quantity $d$. Then, $\delta_{\text{add}}$ would be equal to the moving average SFS value between current state $s_t$ and past state $s_{t - d}$. We can use a similar approach for $\delta_{\text{local}}$, but instead select $d$ to be the desired maximum number of actions away from a landmark we would want to be while still being localized to it.

---

### Decision · Program_Chairs · 2021-09-27

**Decision:**

Accept (Poster)

**Comment:**

All reviewers agree that this is a strong, clearly described empirical work. Although it uses mostly pre-existing concepts, it combines them in an effective way, and its well-executed experiments demonstrate impressive results on pixel-based benchmarks. The ablation studies conducted as part of the discussion with reviewers will further strengthen this paper.